# Proximal and distal spinal neurons innervating multiple synergist and antagonist motor pools

**Remi Ronzano[1][†], Camille Lancelin[1][†], Gardave Singh Bhumbra[2], Robert M Brownstone[1]\*, Marco Beato[3]\***

[1]Department of Neuromuscular Diseases, University College London, London, United Kingdom; [2]Department of Neuroscience Physiology and Pharmacology, University College London, London, United Kingdom; [3]Neuroscience, Physiology and Pharmacology, University College London, London, United Kingdom

**Abstract** Motoneurons (MNs) control muscle contractions, and their recruitment by premotor circuits is tuned to produce accurate motor behaviours. To understand how these circuits coordinate movement across and between joints, it is necessary to understand whether spinal neurons pre-synaptic to motor pools have divergent projections to more than one MN population. Here, we used modified rabies virus tracing in mice to investigate premotor interneurons projecting to synergist flexor or extensor MNs, as well as those projecting to antagonist pairs of muscles controlling the ankle joint. We show that similar proportions of premotor neurons diverge to synergist and antagonist motor pools. Divergent premotor neurons were seen throughout the spinal cord, with decreasing numbers but increasing proportion with distance from the hindlimb enlargement. In the cervical cord, divergent long descending propriospinal neurons were found in contralateral lamina VIII, had large somata, were neither glycinergic, nor cholinergic, and projected to both lumbar and cervical MNs. We conclude that distributed spinal premotor neurons coordinate activity across multiple motor pools and that there are spinal neurons mediating co-contraction of antagonist muscles.

**\*For correspondence:**
r.brownstone@ucl.ac.uk (RMB);
m.beato@ucl.ac.uk (MB)

[†]These authors contributed equally to this work

**Competing interest:** The authors declare that no competing interests exist.

## Editor's evaluation

This manuscript uses viral tracing to identify interneurons, throughout the spinal cord, which synapse onto motoneurons innervating pairs of flexor and extensor hindlimb muscles. Importantly, the data identifies single premotor interneurons which travel to, and presumably regulate the activity of, multiple motor pools. It is possible that these premotor neurons are involved in regulating muscle stiffness across a joint.

## Introduction

The spinal cord is ultimately responsible for organizing movement by controlling the activation pattern of motoneurons (MNs), which in turn produce appropriate patterns of muscle contractions to produce limb movement. Across any single limb joint, there are fundamentally three types of control – or three 'syllables of movement' – possible. The three basic syllables are: (1) changing a joint angle, (2) stiffening a joint, and (3) relaxing a joint. The concatenation of these syllables across joints within and between limbs ultimately produces behaviour (*Brownstone, 2020*; *Wiltschko et al., 2015*).

To change a joint angle, MNs innervating synergist muscle fibres are activated whilst those that innervate antagonist muscle fibres are inhibited. This 'reciprocal inhibition' (*Eccles, 1969*; *Eccles*

**eLife digest** We are able to walk, run and move our bodies in other ways thanks to circuits of neurons in the spinal cord that control how and when our muscles contract and relax. Neurons known as premotor neurons receive information from other parts of the central nervous system and control the activities of groups (known as pools) of motor neurons that directly activate individual muscles.

To bend a joint or move our limbs, the movement of different muscles needs to be coordinated. Previous studies have focused on how premotor neurons activate a pool of motor neurons to contract a single muscle, but it remains unclear if and how some of these premotor neurons can co-activate different pools of motor neurons to control more than one muscle at the same time. Here, Ronzano, Lancelin et al. injected mice with modified rabies viruses labelled with different fluorescent markers to build a map of the premotor neurons that connect to motor neurons controlling the leg muscles.

The experiments revealed that many of the individual premotor neurons in the spinal cords of mice connected to different pools of motor neurons. In the upper region of the spinal cord – which is primarily responsible for controlling the front legs – some large premotor neurons activated motor neurons in this region as well as other motor neurons in a lower region of the spinal cord that controls the back legs. This suggests that these large premotor neurons may be important for coordinating muscles contraction within and between limbs.

Many neurological diseases are associated with difficulties in contracting or relaxing muscles. For example, individuals with a condition called dystonia experience disorganized and excessive muscle contractions that prevent them from being able to bend and straighten their joints properly. By helping us to understand how the body coordinates the activities of multiple limbs at the same time, the findings of Ronzano, Lancelin et al. may lead to new lines of research that ultimately improve the quality of life of patients with dystonia and other similar neurological diseases.

*et al., 1956*) is mediated locally by spinal interneurons (INs) throughout the spinal cord; this syllable has been fairly well characterized, with responsible neurons identified and classified (*Alvarez et al., 2005*; *Benito-Gonzalez and Alvarez, 2012*; *Sapir et al., 2004*; *Zhang et al., 2014*).

The other two syllables are less well studied, but it is clear that behavioural joint stiffening requires co-activation of MNs innervating antagonist muscle groups, while joint relaxation would require co-inhibition of these MNs. Co-contraction has largely been thought to result from brain activity (*Humphrey and Reed, 1983*), whereas circuits mediating co-inhibition remain elusive. Since the spinal cord controls movement not only across single joints but throughout the body, it is natural to consider whether it contains the circuits necessary to produce these different syllables.

To identify whether these syllables are produced by spinal circuits, several questions can be asked: Does the spinal cord contain circuits that lead to co-activation or co-inhibition of different pools of MNs – either synergists or antagonists? Does each motor pool have its own dedicated population of premotor INs, and are these INs interconnected in such a way that they can produce contraction of different muscle groups? Or are there populations of INs that project to multiple motor pools in order to effect contraction (or relaxation) of multiple muscles? Indeed, INs that have activity in keeping with innervation of multiple synergists, leading to motor 'primitives' or synergies (*Bizzi and Cheung, 2013*; *Giszter, 2015*; *Hart and Giszter, 2010*; *Takei et al., 2017*; *Tresch and Jarc, 2009*) have been identified, but knowledge of their locations and identities remains scant.

Normal behaviours in quadrupeds as well as bipeds require coordination of syllables across joints between forelimbs and hindlimbs. This coordination relies on populations of propriospinal neurons projecting in either direction between the lumbar and cervical enlargements (*Eidelberg et al., 1980*; *Giovanelli Barilari and Kuypers, 1969*; *Miller and van der Meché, 1976*; *Ruder et al., 2016*). Long descending propriospinal neurons (LDPNs) were first proposed in cats and dogs more than a century ago (*Sherrington and Laslett, 1903*), and their existence has been confirmed in several other species including humans (*Alstermark et al., 1987a*; *Alstermark et al., 1987b*; *Ballion et al., 2001*; *Brockett et al., 2013*; *Flynn et al., 2017*; *Giovanelli Barilari and Kuypers, 1969*; *Jankowska et al., 1974*; *Mitchell et al., 2016*; *Nathan et al., 1996*; *Ni et al., 2014*; *Reed et al., 2009*; *Ruder et al., 2016*; *Skinner et al., 1979*). While LDPNs that establish disynaptic connections to lumbar MNs have been identified, it was initially suggested that at least some cervical LDPNs could establish monosynaptic

inputs to lumbar MNs (*Jankowska et al., 1974*). This connectivity was later confirmed using monosynaptic modified rabies virus (RabV) tracing (*Ni et al., 2014*). More recently, descending and ascending spinal neurons and their involvement in the control of stability and interlimb coordination have been characterized, but these studies did not directly focus on monosynaptic premotor circuits (*Pocratsky et al., 2017*; *Ruder et al., 2016*). It is likely that LDPNs function to ensure coordination between forelimbs and hindlimbs, and they could be an important source of premotor input to MNs, providing a substrate for coordination between distant joints.

In the present study, we examine circuits underlying co-activation and co-inhibition in the spinal cord by assessing premotor neurons through the use of RabV tracing techniques (*Ronzano et al., 2021*; *Ugolini, 1995*; *Wickersham et al., 2007*). We used glycoprotein (G)-deleted RabV (ΔG-Rab), and supplied G to MNs through crossing ChAT-Cre mice with RΦGT mice (*Ronzano et al., 2021*; *Takatoh et al., 2013*). We injected ΔG-RabV tagged with two different fluorescent proteins into hindlimb muscle pairs of ChAT-Cre mice to retrogradely trace premotor circuits throughout the spinal cord. At the lumbar level, this method revealed apparent low rates of INs projecting to both MN pools targeted. As the distance from targeted MN pool to premotor INs increased, the density of infected premotor INs decreased. But the apparent rate of divergence to multiple pools was higher in thoracic and cervical regions than in the lumbar spinal cord. Interestingly, the extent of divergence throughout the spinal cord was similar whether injections were performed in flexor or extensor pairs, or in synergist or antagonist pairs of muscles. In addition, a population of premotor LDPNs was identified in the cervical spinal cord. These neurons had a high rate of divergence and large somata, projected contralaterally, were neither glycinergic nor cholinergic, located in lamina VIII, and projected to cervical MNs as well as lumbar MNs. Together, these data show that the spinal cord contains premotor INs that project to multiple motor pools (including antagonists), and could thus form substrates for the fundamental syllables of movement.

## Results

### Lumbar premotor INs reveal similar divergence patterns to synergist and antagonist motor pools

Given evidence that INs are involved in motor synergies (*Hart and Giszter, 2010*; *Levine et al., 2014*; *Takei et al., 2017*; *Takei and Seki, 2010*), we would expect that there would be INs in the lumbar spinal cord that project to synergist motor pools. We thus first investigated whether such premotor INs could be infected with two RabVs expressing two different fluorescent proteins injected in pairs of muscles. We injected ΔG-Rab expressing eGFP or mCherry into synergist ankle extensors (lateral gastrocnemius [LG] and medial gastrocnemius [MG]) or synergist ankle flexors (tibialis anterior [TA] and peroneus longus [PL]) in ChAT-Cre;RΦGT P1-P3 mice (*Ronzano et al., 2021*). These mice selectively express rabies G in cholinergic neurons (including MNs), providing the necessary glycoprotein for retrograde trans-synaptic transfer from infected MNs to premotor INs (*Figure 1A*). After 9 days, we visualized the distribution of premotor INs that expressed one or both fluorescent proteins, specifying the premotor INs that make synaptic contact with two motor pools as 'divergent' premotor INs (*Figure 1—figure supplement 1A, B*, *Figure 1—figure supplement 2A, B*). We found divergent premotor INs distributed across the lumbar spinal cord (*Figure 1—figure supplement 4A, B*) bilaterally in the ventral quadrants and ipsilaterally in the dorsal quadrant of the spinal cord (*Figure 1C, D*), consistently across experiments (*Figure 1—figure supplement 5A*, *Supplementary file 1*). Across the lumbar spinal cord, we quantified infected MNs and found that 380 MNs were labelled from synergist injections (*n* = 4, two extensor and two flexor pairs). Notably, five MNs were double labelled, most likely due to secondary infection of synaptically connected MNs (*Supplementary file 1*, *Bhumbra and Beato, 2018*). We then quantified premotor INs on one of every three sections and found that 4.0% ± 0.3 % (276/7043, *n* = 4, two extensor and two flexor pairs, *Figure 1—figure supplement 1C, D*, and *Figure 1—figure supplement 2C, D*) of labelled premotor INs were double labelled, confirming that INs can be infected by more than one RabV. We would expect this to be an underestimate of the number of INs that have divergent projections since RabV is not expected to label 100 % of presynaptic neurons and as there is a reduced efficiency of double infections compared to single infections (*Ohara et al., 2009*; see Discussion).

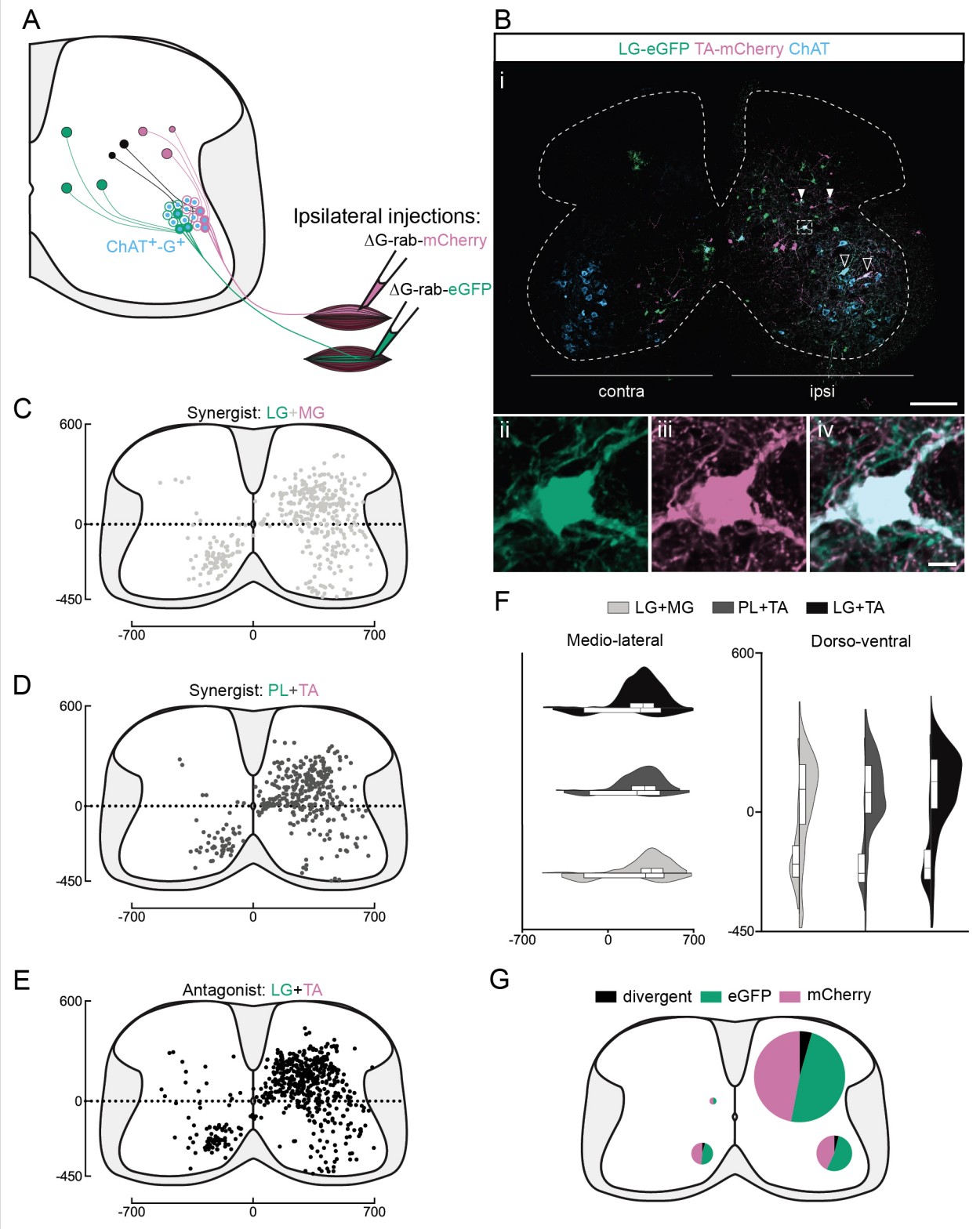

**Figure 1.** Organization of divergent premotor interneurons (INs) in the lumbar spinal cord. (**A**) Experimental strategy to describe divergent premotor INs that project to two motor pools of synergist (injection in tibialis anterior [TA] and peroneus longus [PL] or lateral gastrocnemius [LG] and medial gastrocnemius [MG]) or antagonist (TA and LG) pair of muscles. (**Bi**) Representative example of a lumbar transverse section following an injection in the TA (ΔG-Rab-mCherry) and LG (ΔG-Rab-eGFP), showing ChAT (grey blue), GFP (Green Fluorescent Protein, green), and mCherry (pink). A divergent premotor IN is highlighted in the dashed box. Filled arrowheads show divergent premotor INs and contour arrowheads show infected motoneurons

*Figure 1 continued on next page*

*Figure 1 continued*

(MNs). The dashed line drawn outlines the grey matter contour. Higher magnification of a divergent premotor IN that has been infected by the ΔG-Rab-eGFP and ΔG-Rab-mCherry, showing (**ii**) eGFP, (**iii**) mCherry, and (**iv**) the overlay. More representative examples of lumbar sections following injections in LG and MG, TA and PL, and LG and TA are shown in *Figure 1—figure supplements 1–3*, respectively. Distribution of the lumbar divergent premotor INs following injections in (**C**) LG and MG (*n* = 2), (**D**) PL and TA (*n* = 2), and (**E**) LG and TA (*n* = 3). (**F**) Asymmetric violin plots showing the medio-lateral and dorso-ventral distributions of divergent premotor INs. The halves correspond, respectively, to the dorsal (top) and ventral (bottom) distributions and to the ipsilateral (right) and contralateral (left) distributions of divergent premotor INs in the lumbar cord. Violin areas were normalized on the number of divergent INs. (**G**) Distribution of the premotor INs within each quadrant of the lumbar cord, with pie sizes proportional to the percentage of premotor INs in each quadrant of the lumbar cord. Numbers along the axis indicate distances (in μm). Scale bars: (Bi) 200 μm; (Biv) 10 μm. Raw number of eGFP, mCherry, and double-labelled premotor neurons per samples per muscle pair injected, is shown in *Figure 1—figure supplements 1–3*.

The online version of this article includes the following source data and figure supplement(s) for figure 1:

**Source data 1.** Source data for *Figure 1C-G*.

**Figure supplement 1.** Divergent premotor interneurons (INs) in the lumbar spinal cord following injections in synergists lateral gastrocnemius (LG) and medial gastrocnemius (MG).

**Figure supplement 2.** Divergent premotor interneurons (INs) in the lumbar spinal cord following injections in synergists peroneus longus (PL) and tibialis anterior (TA).

**Figure supplement 3.** Divergent premotor interneurons (INs) in the lumbar spinal cord following injections in antagonists lateral gastrocnemius (LG) and tibialis anterior (TA).

**Figure supplement 4.** Rostro-caudal distributions of divergent lumbar premotor interneurons (INs).

**Figure supplement 4—source data 1.** Source data for *Figure 1—figure supplement 4A-C*.

**Figure supplement 5.** Medio-lateral and dorso-ventral distributions of divergent premotor neurons across individual experiments.

**Figure supplement 6.** Excitatory boutons from infected premotor neurons and apposed to motoneurons (MNs) reveal divergence through different segments and regions of the spinal cord.

We next sought to determine whether this divergence was restricted to synergist motor pools or whether there are also premotor INs that diverge to antagonist pools and could thus be involved in co-contraction or joint stiffening. Following injections into flexor (TA) and extensor (LG) muscles, 260 MNs were labelled (*n* = 3 antagonist pairs), one of them being double labelled (*Supplementary file 1*). Following these injections, we also found divergent INs (*Figure 1B*, *Figure 1—figure supplement 3A, B*). We found a similar rate of divergence to antagonist pools as to synergist muscles, with 4.7% ± 0.5 % (206/4341, *n* = 3 antagonist pairs, *Figure 1E*, *Figure 1—figure supplement 3C–E*) double labelled. The mapping of all divergent INs in every section revealed that, whether injections were in synergist (*n* = 4, two extensor and two flexor synergist pairs) or antagonist (*n* = 3 pairs) pairs of muscles, double-labelled premotor INs were distributed similarly (*Figure 1F, G*, *Figure 1—figure supplement 4*, *Figure 1—figure supplement 5A*, and *Supplementary file 1* for summary of individual experiments). The proportion of divergent cells was calculated from the ratio of double and single infected cells in 1/3 sections, in order to avoid double counting cells present in consecutive sections (see Methods). Equal proportions of divergent premotor INs were found in the ventral ipsilateral quadrant synergists: 74/1913 (3.9%) vs antagonists: (46/1046 [4.4%]), ventral contralateral quadrant (46/1020 [3.8%] vs 21/502 [4.2%]), and dorsal ipsilateral quadrant (153/3874 [3.9%] vs 134/2651 [5.1%]). There were few labelled neurons in the dorsal contralateral quadrant following either synergist or antagonist injections and a similarly low proportion were double labelled (in 1/3 sections: 3/236 [1.3%] and 5/142 [3.5%], respectively). Divergence in premotor circuits is thus common, with at least 1/25 (see Discussion) premotor INs diverging to two MN pools, whether synergists or antagonists.

Since motor synergies can span across more than a single joint, it is possible that divergent premotor INs could project to motor pools other than those injected. Indeed, following injection of ΔG-Rab-mCherry into the TA muscle, we could visualize mCherry-positive excitatory (vGluT2+) boutons in apposition to L1 (*Figure 1—figure supplement 6A, D–E*), as well as to thoracic (as rostral as at least T10) MNs (*Figure 1—figure supplement 6B, C*), that is, three to seven segments rostral to the infected motor pool. mCherry-positive excitatory boutons on MNs were consistently observed in all upper lumbar and thoracic sections taken from three injected mice (three to four sections in each region). This observation, in agreement with a previous study that described premotor INs coordinating the activity of multiple lumbar motor groups from L2 to L5 (*Levine et al., 2014*), supports the possibility that thoraco-lumbar premotor circuits comprise a substrate for multi-joint synergies.

## Thoracic premotor neurons project to multiple lumbar motor pools

In order to maintain posture and stability, trunk muscles are coordinated with hindlimb movements. Neurons in the thoracic cord that are premotor to lumbar MNs have previously been described (*Ni et al., 2014*); we thus next examined the projections of thoracic premotor neurons to lumbar motor pools. These premotor neurons were found with decreasing density from T11 through T3 whether the injections were in extensor (LG and MG; *Figure 2—figure supplement 1A, B*) or flexor (TA and PL, Figure 2A, *Figure 2—figure supplement 2A-B*) pairs of muscles (*Figure 2—figure supplement 4A, B*). The distributions of single labelled as well as divergent premotor neurons were similar whether injections were performed in flexor, extensor, or antagonist pairs of muscles (*Figure 2B–H*, *Figure 2—figure supplement 4A–C*, *Figure 1—figure supplement 5B*). Divergence rates calculated from the whole thoracic spinal cords were similar between synergist and antagonist injections with 16.2% ± 5.7% (77/497, *n* = 4, 2 extensor and two flexor pairs, *Figure 2B, C*, *Figure 2—figure supplement 1C, D*, *Figure 2—figure supplement 2C, D*) and 9.0% ± 0.7% (59/401, *n* = 3 antagonist pairs, *Figure 2D*, *Figure 2—figure supplement 3C, E*), respectively. In all animals (7/7), the overall proportion of double-labelled neurons in the thoracic spinal cord was higher than in the lumbar cord (13.1% ± 5.6%, Figure 7B).

In all animals (7/7), most divergent premotor neurons in the thoracic cord were located in the ipsilateral dorsal quadrant (46/77, *n* = 4 synergist and 42/59, *n* = 3 antagonist pairs, *Figure 2E–H*), and within this quadrant 22.1% ± 8.6% (46/188 synergists and 42/211 antagonists, *Figure 2E, F*) of premotor neurons were double labelled. The divergence rates in the two ventral quadrants were lower: in the ventral cord, double-labelled neurons were observed in 5/7 animals (3/4 synergist; 2/3 antagonist in both quadrants) ipsilaterally (6.7% ± 4.8%; 10/118 synergist and 5/70 antagonist pairs), as well as contralaterally (11.1% ± 10.7%; 21/167 synergist and 12/104 antagonist pairs) to the injection (*Figure 2E, F*). Thus, there are premotor neurons throughout the thoracic cord that project directly to more than one motor pool, including antagonist pairs, in the lumbar spinal cord, with most of these located in the ipsilateral dorsal quadrant.

## Cervical premotor long propriospinal descending neurons diverge and share a typical location and morphology

Cervical long descending propriospinal neurons (LDPNs) have been shown to modulate interlimb coordination to provide stability (*Eidelberg et al., 1980*; *Miller and van der Meché, 1976*; *Pocratsky et al., 2017*; *Ruder et al., 2016*). Given that cervical premotor LDPNs projecting to TA MNs have previously been demonstrated (*Ni et al., 2014*), we asked whether these neurons could be premotor to hindlimb and/or hindlimb–forelimbs MN pairs.

We found that premotor LDPNs projecting to flexor (TA and PL) and extensor (LG and MG) MNs were localized throughout the rostro-caudal extent of the ventral cervical cord with an enrichment between C6 and T1 (*Figure 3—figure supplement 4*). Of 92 premotor LDPNs, 88 were localized in the ventral quadrants, 68 of which were in contralateral lamina VIII (*n* = 7, four synergist and three antagonist pairs, *Figure 3A–F*, *Figure 3—figure supplement 4*). A substantial proportion of premotor LDPNs was double labelled, with the proportion and location of double labelling similar across experiments (*Figure 1—figure supplement 5C* and *Supplementary file 1*) whether injections were into synergist or antagonist pairs (42.4% ± 22.1 % per animal, total of 19/55 neurons, *n* = 4 synergist pairs and 47.9% ± 7.1 % per animal, total of 19/37 neurons, *n* = 3 antagonist pairs, *Figure 3E, F*, *Figure 3—figure supplement 1*, *Figure 3—figure supplement 2*, *Figure 3—figure supplement 3*). This apparent divergence rate of LDPNs in the cervical cord was higher than in the lumbar and thoracic cords in all animals (7/7, Figure 7B). These divergent premotor LDPNs exhibited a stereotypical morphology with an unusually large soma (774 ± 231 μm$^2$, *n* = 38 premotor LDPNs) compared to the double-labelled premotor neurons in the thoracic and lumbar cords (respectively, 359 ± 144 and 320 ± 114 μm$^2$, *n* = 135 premotor neurons [thoracic], *n* = 61 premotor INs [lumbar], p < 0.0001 Kruskal–Wallis test, p < 0.0001 [lumbar vs cervical], and p < 0.0001 [thoracic vs cervical], Dunn's multiple comparisons test). On average, the cross-sectional area of divergent cervical LDPNs was comparable to that of cervical MNs (661 ± 86 μm$^2$, *n* = 17 MNs, *Figure 3H*). Their location and size suggest that these divergent, commissural cervical premotor LDPNs may constitute a somewhat homogenous population.

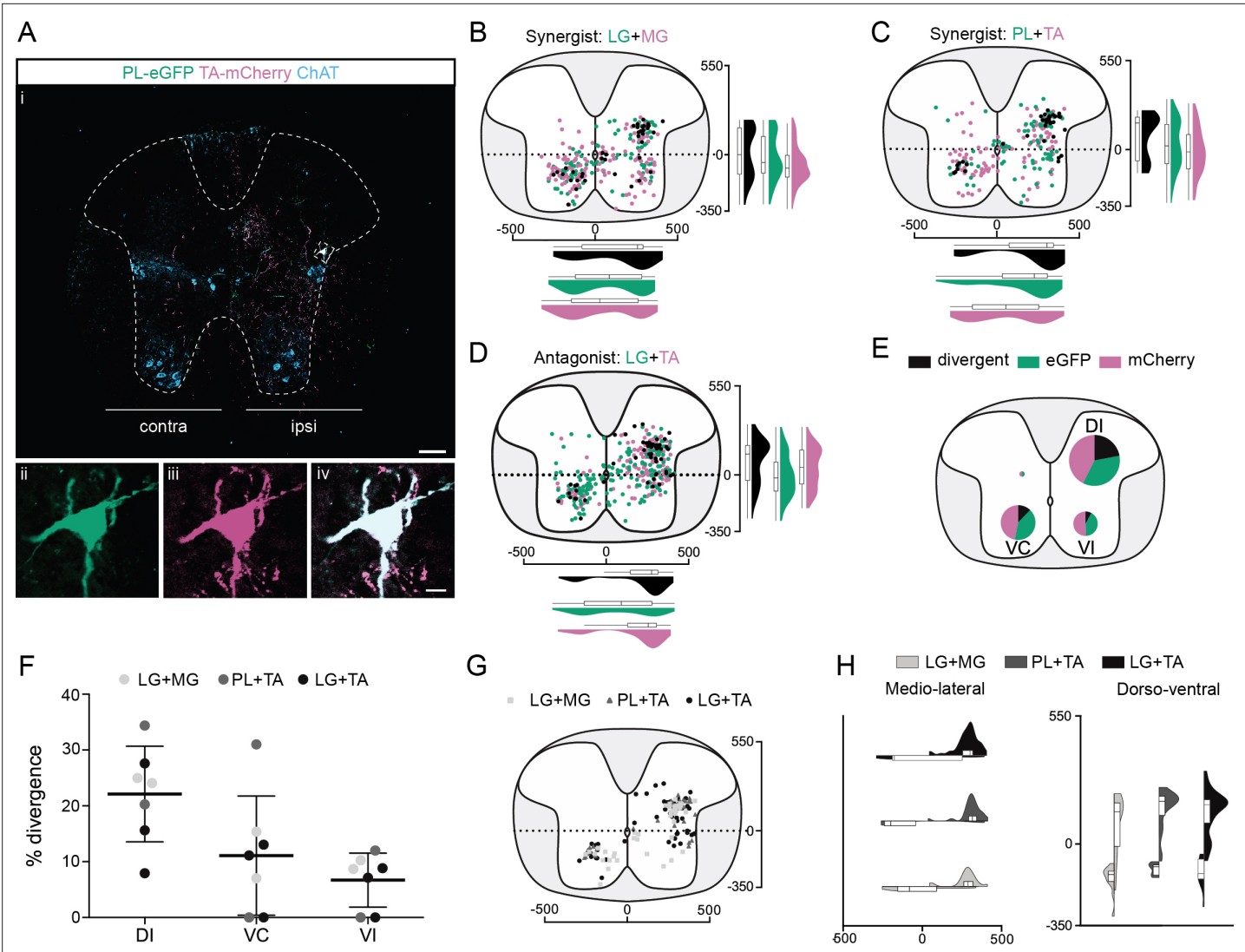

**Figure 2.** Organization of divergent premotor neurons in the thoracic segments. (**Ai**) Representative example of a thoracic transverse section following an injection in the peroneus longus (PL) (ΔG-Rab-eGFP) and tibialis anterior (TA) (ΔG-Rab-mCherry), showing ChAT (grey blue), GFP (green), and mCherry (pink). A divergent premotor neuron is highlighted in the dashed box. The dashed line drawn outlines the grey matter contour. Higher magnification of a divergent premotor neuron that has been infected by both ΔG-Rab-eGFP and ΔG-Rab-mCherry, showing (**ii**) eGFP, (**iii**) mCherry, and (**iv**) the overlay. More representative examples of thoracic sections following injections in lateral gastrocnemius (LG) and medial gastrocnemius (MG), TA and PL, and LG and TA are shown in **Figure 2—figure supplements 1–3**, respectively. Distribution of the thoracic premotor neurons infected following injections in (**B**) LG and MG (*n* = 2), (**C**) PL and TA (*n* = 2), and (**D**) LG and TA (*n* = 3). Divergent premotor neurons infected from both injections are labelled in black. The violin plots show the dorso-ventral and medio-lateral distributions of divergent (black), GFP-positive (green), and mCherry-positive (pink) premotor neurons along the medio-lateral and dorso-ventral axis. Each violin area is normalized to 1. (**E**) Pies showing the distribution of infected premotor neurons in each quadrant; the size of the pies is proportional to the number of infected neurons. (**F**) Plot showing the divergence rate in each quadrant of the thoracic cord. DI: dorsal ipsilateral; VC: ventral contralateral; VI: ventral ipsilateral. (**G**) Overlap of distributions of divergent thoracic premotor neurons followings each pair of muscles injected. (**H**) Asymmetric violin plots showing the medio-lateral and dorso-ventral distributions of divergent premotor neurons. The halves correspond, respectively, to the dorsal (top) and ventral (bottom) distributions and to the ipsilateral (right) and contralateral (left) distributions of divergent premotor neurons in the thoracic cord. Violin areas were normalized on the number of divergent neurons. When not specified numbers along the axis indicate distances (in μm). Scale bars: (Ai) 100 μm; (Aiv) 10 μm. Raw number of eGFP, mCherry, and double-labelled premotor neurons per samples per muscle pair injected, is shown in **Figure 2—figure supplements 1–3**.

The online version of this article includes the following figure supplement(s) for figure 2:

**Source data 1.** Source data for **Figure 2B–G**.

**Figure supplement 1.** Divergent premotor interneurons (INs) in the thoracic spinal cord following injections in synergists lateral gastrocnemius (LG) and medial gastrocnemius (MG).

*Figure 2 continued on next page*

*Figure 2 continued*

**Figure supplement 2.** Divergent premotor interneurons (INs) in the thoracic spinal cord following injections in synergists peroneus longus (PL) and tibialis anterior (TA).

**Figure supplement 3.** Divergent premotor interneurons (INs) in the thoracic spinal cord following injections in antagonists lateral gastrocnemius (LG) and tibialis anterior (TA).

**Figure supplement 4.** Rostro-caudal distributions of divergent thoracic premotor neurons.

**Figure supplement 4—source data 1.** Source data for *Figure 2—figure supplement 4A–C*.

## Cervical premotor LDPNs are neither glycinergic nor cholinergic

To determine the neurotransmitter phenotype of the premotor LDPNs, we used single ΔG-Rab-mCherry injections in ChAT-Cre;RΦGT mice crossed with mice expressing eGFP under the control of the promoter for the neuronal glycine transporter GlyT2 (*Figure 4A*, *Zeilhofer et al., 2005*). GlyT2 is expressed in the vast majority of spinal inhibitory INs (*Todd et al., 1996*; *Todd and Sullivan, 1990*), making GlyT2-eGFP mice a suitable tool to determine whether premotor LDPNs are inhibitory. Given that at least 40 % of the labelled INs in the cervical region are divergent (see above), many of the neurons labelled following even single RabV injections would be expected to be divergent. Following injection into LG (*Figure 4A*), we found that only 1/21 infected cervical commissural premotor LDPNs was eGFP positive (*n* = 3 LG injections, *Figure 4B, C and F*). Since none of the labelled neurons expressed ChAT, the majority of cervical premotor LDPNs are likely to be glutamatergic by exclusion. However, in agreement with previous results from TA injections (*Ni et al., 2014*), single-labelled thoracic premotor neurons comprised a mixed population of inhibitory and non-inhibitory neurons (34.4% ± 5.9%, 96/273, mCherry+ eGFP + premotor neurons, *n* = 3 LG injections, *Figure 4D–F*). We cannot determine whether the thoracic or lumbar GFP+ or GFP− premotor INs are divergent, as these data were obtained following single injections. However, in the lumbar cord, as expected, we observed that some divergent INs were cholinergic (*Figure 4—figure supplement 1*).

## A subset of cervical premotor LDPNs arise from the V0 or dI2 domain

We next sought to determine the genetic provenance of divergent cervical LDPNs. Among the classes of ventral INs defined by the early expression of transcription factors (*Lee and Pfaff, 2001*), the V0 and V3 cardinal classes are known to project to contralateral MNs. These classes can be further subdivided, with all V3 subclasses being glutamatergic (*Zhang et al., 2008*), and V0 INs being neuromodulatory $V0_C$, cholinergic (*Miles et al., 2007*), inhibitory $V0_D$, dorsal (*Talpalar et al., 2013*), or excitatory $V0_V$, ventral (*Talpalar et al., 2013*), or $V0_G$, medial glutamatergic neurons that project to dorsal and intermediate lamina but not to MNs (*Zagoraiou et al., 2009*). Since previous studies showed that none of the LDPNs with soma in the cervical cord belong to the V3 population (*Flynn et al., 2017*), we sought to determine whether these LDPNs were of the V0 class.

V0 INs are defined by their embryonic expression of the transcription factor Dbx1 (*Pierani et al., 2001*) and Evx1 (*Moran-Rivard et al., 2001*). However, neither of these two transcription factors can reliably be detected at the postnatal ages of our mice. On the other hand, Lhx1 is expressed throughout the V0 and V1 populations (as well as dI2, dI4, and $dIL_A$ populations) and may be detectable at this early postnatal stage (*Skarlatou et al., 2020*). However, V1 and $V0_D$ INs are glycinergic (*Alvarez et al., 2005*; *Talpalar et al., 2013*), $V0_C$ are cholinergic (*Miles et al., 2007*). Since we have shown that LDPNs are negative for GlyT2 and ChAT and dI4 and $dIL_A$ INs are dorsal neurons (*Glasgow et al., 2005*; *Pillai et al., 2007*) expression of Lhx1 would point to cervical LDPNs belonging to either the $V0_V$ or dI2 class. In fact, it has very recently been shown that dorsally derived excitatory dI2 INs migrate to this region in the chick spinal cord and have divergent axons along the length of the cord and to the cerebellum (*Haimson et al., 2021*). While these neurons are not premotor in the chick (*Haimson et al., 2021*), it is possible that they are in the mouse.

Following injection of gastrocnemius (GS, *n* = 4, *Figure 5A*), we detected 33 premotor LDPNs. Of these infected cervical premotor LDPNs, 8 (~24%) were clearly Lhx1 positive (*Figure 5B, C*). Given that there is a decrease of Lhx1 expression along the course of postnatal development (*Figure 5—figure supplement 1*), it is possible that the proportion of premotor LDPNs that were positive for Lhx1 was underestimated. Nevertheless, although we cannot conclude that the identified LDPNs arise

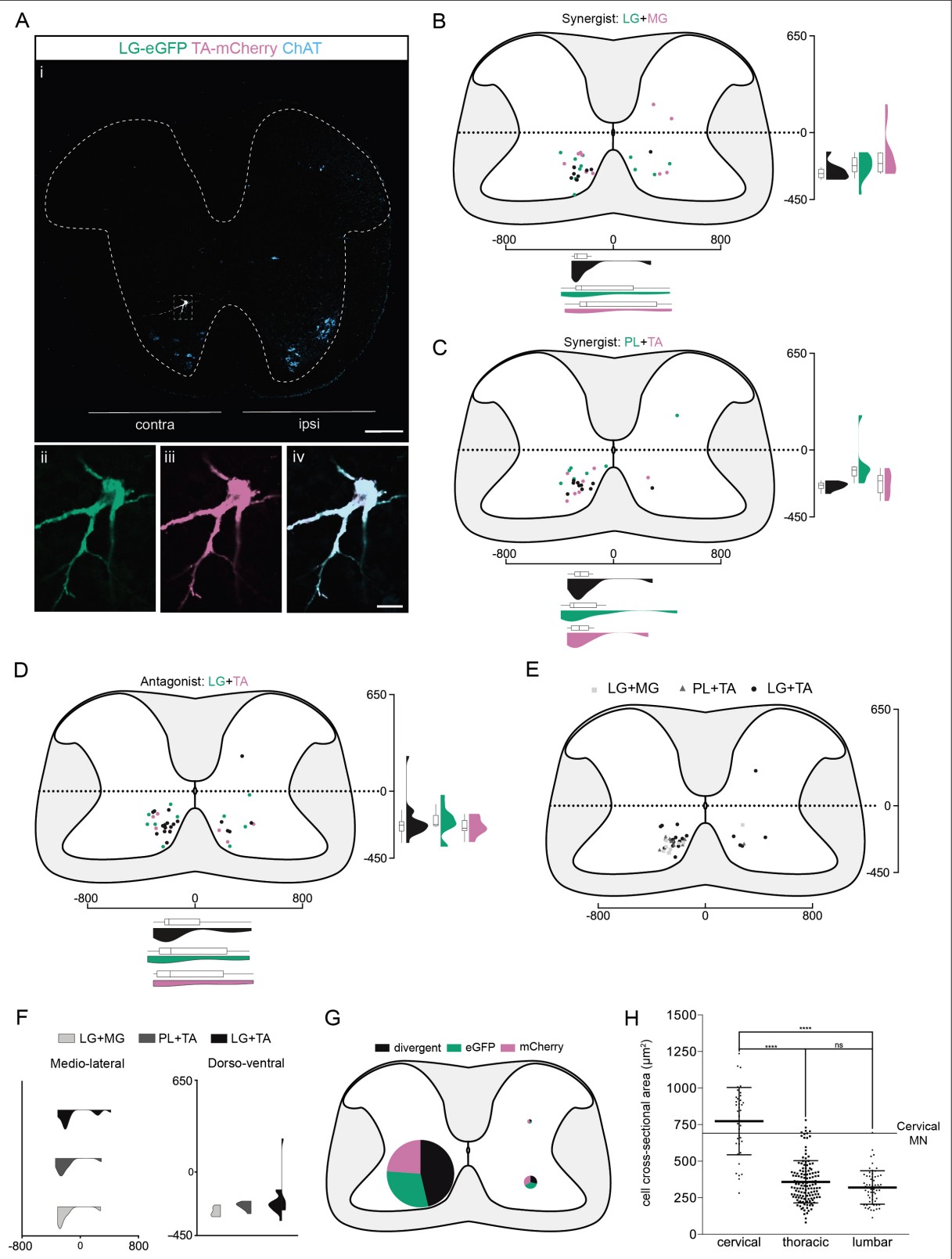

**Figure 3.** Organization of divergent premotor long descending propriospinal neurons (LDPNs) in the cervical spinal cord. (**Ai**) Representative example of an upper cervical transverse section following an injection in the lateral gastrocnemius (LG) (ΔG-Rab-eGFP) and tibialis anterior (TA) (ΔG-Rab-mCherry), showing ChAT (grey blue), GFP (green), and mCherry (pink). A divergent premotor LDPN is highlighted in the dashed box. The dashed line drawn outlines the grey matter contour. Higher magnification of the divergent premotor LDPN, showing (**ii**) eGFP, (**iii**) mCherry, and (**iv**) the overlay. More

*Figure 3 continued on next page*

*Figure 3 continued*

representative examples of cervical sections following injections in LG and medial gastrocnemius (MG), TA and peroneus longus (PL), and LG and TA are shown in *Figure 3—figure supplements 1–3*, respectively. (**B–D**) Distribution of the cervical premotor LDPNs following injections in (**C**) LG and MG (n = 2), (**D**) PL and TA (n = 2), and (**E**) LG and TA (n = 3). Divergent premotor LDPNs infected from both injections are labelled in black. The violin plots show the dorso-ventral and medio-lateral distributions of divergent (black), GFP-positive (green), and mCherry-positive (pink) premotor LDPNs along the medio-lateral and dorso-ventral axis. Each violin area is normalized to 1. (**E**) Overlap of the distribution of cervical divergent premotor LDPNs followings each pair of muscles injected. (**F**) Asymmetric violin plots showing the medio-lateral and dorso-ventral distributions of premotor divergent LDPNs. The halves correspond, respectively, to the dorsal (top) and ventral (bottom) distributions and to the ipsilateral (right) and contralateral (left) distributions of divergent premotor LDPNs in the cervical cord. Violin areas were normalized on the number of divergent neurons. (**G**) Pies showing the distribution of infected premotor LDPNs in each quadrant; the size of the pies is proportional to the number of infected premotor LDPNs in each quadrant. (**H**) Plot showing the distribution of the sectional areas of divergent premotor neurons in each region of the spinal cord. The dashed line (labelled cervical motoneuron [MN]) corresponds to the mean sectional area of cervical MNs (n = 17 MNs). When not specified numbers along the axis indicate distances (in µm). Scale bars: (Ai) 200 µm; (Aiv) 20 µm. Raw number of eGFP, mCherry, and double-labelled premotor neurons per samples per muscle pair injected, are shown in *Figure 3—figure supplements 1–3*.

The online version of this article includes the following figure supplement(s) for figure 3:

**Source data 1.** Source data for *Figure 3B–H*.

**Figure supplement 1.** Divergent premotor interneurons (INs) in the cervical spinal cord following injections in synergists lateral gastrocnemius (LG) and medial gastrocnemius (MG).

**Figure supplement 2.** Divergent premotor interneurons (INs) in the cervical spinal cord following injections in synergists peroneus longus (PL) and tibialis anterior (TA).

**Figure supplement 3.** Divergent premotor interneurons (INs) in the cervical spinal cord following injections in antagonists lateral gastrocnemius (LG) and tibialis anterior (TA).

**Figure supplement 4.** Rostro-caudal distributions of divergent cervical premotor neurons.

**Figure supplement 4—source data 1.** Source data for *Figure 3—figure supplement 4A–C*.

from a homogenous population, it is likely that at least a portion of them arise from $V0_v$ neurons and/or dI2 neurons.

## Cervical premotor LDPNs also project to local cervical MNs

Given that propriospinal neurons are involved in interlimb coordination, we next sought to determine whether the divergent cervical premotor LDPNs also project to cervical MNs. We therefore performed a series of experiments in which we injected forearm muscles (FMs) with ΔG-Rab-mCherry, and extensor hindlimb GS with ΔG-Rab-eGFP.

Since it has been suggested that LDPNs participate in ipsilateral control of forelimb and hindlimb (*Miller and van der Meché, 1976*), we sought to determine if premotor LDPNs project to homolateral lumbar and cervical motor pools (*Figure 6A*). When homolateral limbs were targeted, we found that some premotor LDPNs infected from ankle extensor injections were also infected from homolateral FMs injection (in 5/6 animals, 16/80 premotor LDPNs were also infected from FMs injection 18.7% ± 12.9%, *Figure 6B–D* and *Figure 7*). These divergent premotor LDPNs that projected to lumbar and cervical MNs were all located in the ventral quadrants with 11/16 located in contralateral lamina VIII, and were distributed throughout the rostro-caudal extent of the cervical cord, including segments rostral (C4) to the MN pools innervating the injected forelimb muscles. Furthermore, they had a soma size similar to the premotor LDPNs double labelled by dual hindlimb injections (632 ± 236 µm², p = 0.056, $n_1$ = 16 premotor LDPNs infected from both homolateral forelimb and hindlimb injections vs $n_2$ = 38 divergent premotor LDPNs infected from dual hindlimb injections (see above), Mann–Whitney test; *Figure 6E*).

Given the involvement of LPDNs in the diagonal synchronization of forelimb and hindlimb during locomotion (*Bellardita and Kiehn, 2015*; *Ruder et al., 2016*; *Sherrington et al., 1906*), we also injected contralateral FMs and GS (*Figure 6—figure supplement 1A*). We found that 2/26 cervical premotor LDPNs were also infected from the FMs injection with one divergent LDPNs in the lamina VIII contralateral to the hindlimb injection in each of two of the three injected animals (*Figure 6—figure supplement 1B*). Thus, at least a few cervical premotor LDPNs monosynaptically project to diagonal lumbar and cervical MNs. However, given the paucity of these cervical premotor LDPNs projecting to local cervical MNs, we could not reliably determine whether this subpopulation shared the same morphology as described above.

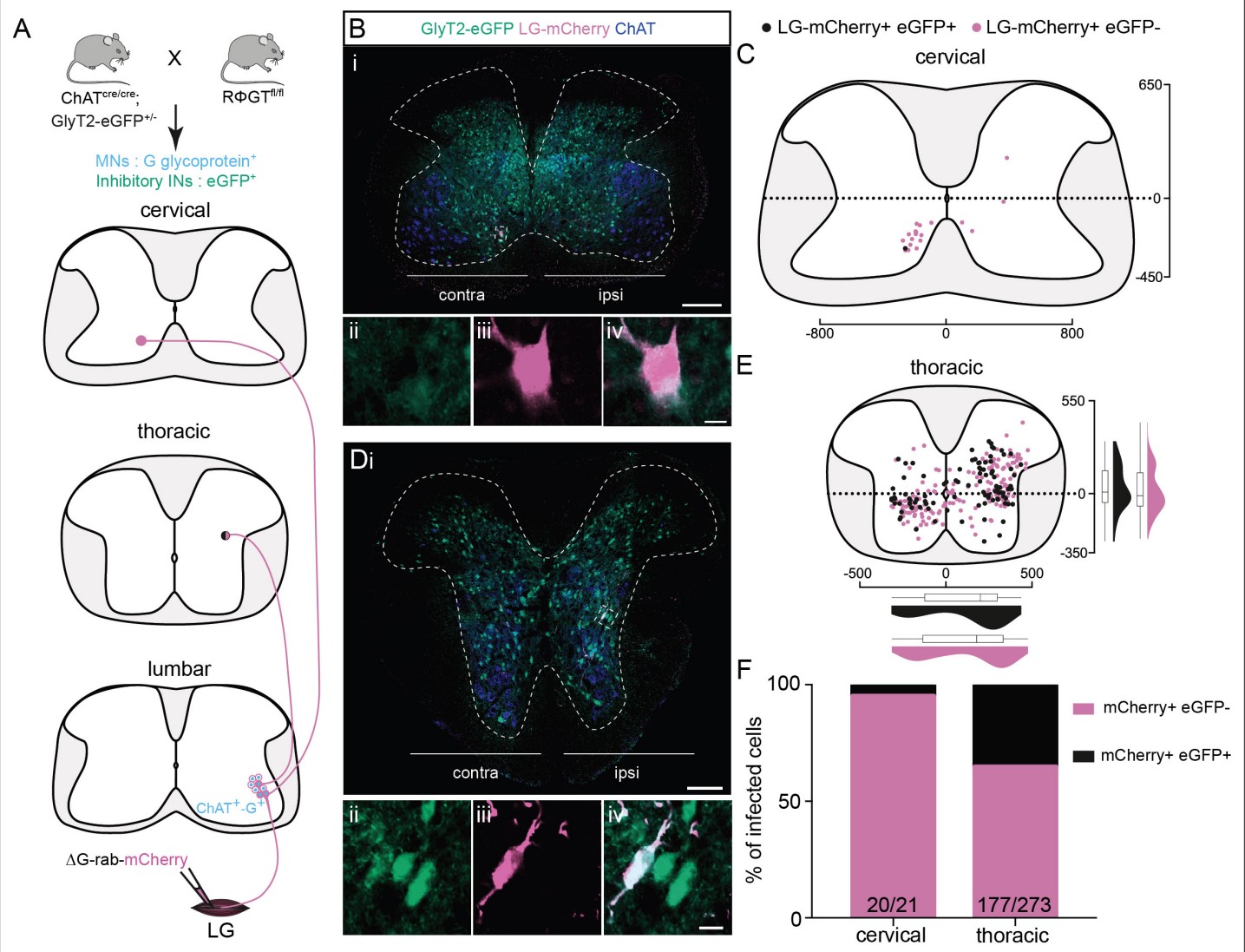

**Figure 4.** Non-glycinergic, non-cholinergic cervical premotor long descending propriospinal neurons (LDPNs), and mixed populations of inhibitory and non-inhibitory thoracic premotor neurons revealed by injections in GlyT2-eGFP; RΦGT mice. (**A**) Experimental strategy to determine whether thoracic and cervical premotor neurons are inhibitory. (**B, D**) Representative example of (Bi) a cervical and (Di) a thoracic transverse section following an injection in the lateral gastrocnemius (LG) (ΔG-Rab-mCherry) using GlyT2-eGFP; RΦGT mice, showing ChAT (blue), GFP (green), and mCherry (pink). The dashed boxes highlight the infected premotor LDPNs. The dashed lines drawn outline the grey matter contours. Higher magnification of the dashed box areas, highlighting (**Bii–iv**) a GFP−, mCherry+ cervical premotor LDPN on the contralateral lamina VIII and (**Dii–iv**) a GFP+, mCherry+ thoracic premotor neuron in ipsilateral intermediate lamina. Distribution of the (**C**) cervical and (**E**) thoracic premotor neurons infected, following injections in the LG of GlyT2-eGFP; RΦGT mice (*n* = 3). The violin plots show the dorso-ventral and medio-lateral distributions of GFP+, mCherry+ (black) and GFP−, mCherry+ (pink) premotor neurons along the medio-lateral and dorso-ventral axis. Each violin area is normalized to 1. (**F**) Proportions of inhibitory premotor neurons in the thoracic and the cervical region of GlyT2-eGFP; RΦGT mice following injections in the LG (*n* = 3). When not specified numbers along the axis indicate distances (in μm). Scale bars: (Bi) 200 μm; (Di) 100 μm; (Biv, Div) 10 μm.

The online version of this article includes the following figure supplement(s) for figure 4:

**Source data 1.** Source data for *Figure 4C, E, F*.

**Figure supplement 1.** Lumbar V0c interneurons (INs) innervate multiple motor pools.

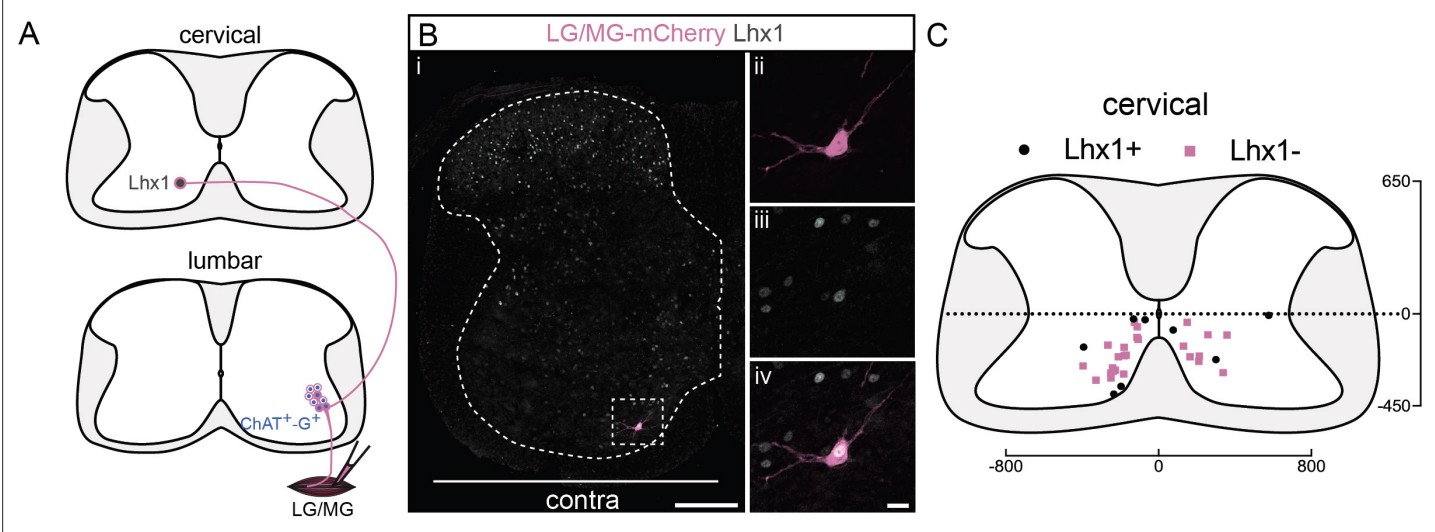

**Figure 5.** A subpopulation of cervical premotor long descending propriospinal neurons (LDPNs) expresses Lhx1. (**A**) Experimental strategy to determine whether cervical premotor LDPNs express Lhx1. (**Bi**) Representative example of a transverse section from the cervical cord following an injection of ΔG-Rab-eGFP in the gastrocnemius (GS) muscles, showing a cervical premotor LDPN infected (pink) expressing Lhx1 (grey). The premotor LDPN expressing Lhx1 is highlighted in the dashed box. The dashed line drawn outlines the grey matter contour. Higher magnification of the premotor LDPN Lhx1+ that has been infected by the ΔG-Rab-mCherry, showing (ii) mCherry, (iii) Lhx1, and (iv) the overlay. (**C**) Distribution of the cervical premotor LDPNs following injections in GS whether they are Lhx1+ (black) or not (pink). Numbers along the axis indicate distances (in µm). Scale bars: (Bi) 200 µm; (Biv) 20 µm. The efficiency of Lhx1 staining along postnatal development is shown *Figure 5—figure supplement 1*.

The online version of this article includes the following figure supplement(s) for figure 5:

**Source data 1.** Source data for *Figure 5C*.

**Figure supplement 1.** The number of neurons labelled with anti-Lhx1 antibody decreases in the spinal cord over postnatal development.

While sharing similar features with the LDPNs infected from dual hindlimb injections, it remains to be determined whether these neurons premotor to hindlimb and forelimb muscles form a homogenous population with the divergent LDPNs.

## Distribution of premotor long ascending propriospinal neurons differs from that of LDPNs

Having identified a population of divergent premotor LDPNs with projections from the cervical to the lumbar region, we next investigated whether ascending propriospinal neurons projecting from the lumbar or thoracic segments to cervical MNs could be identified. Following FMs injections, ascending premotor INs were observed throughout the cord (thoracic to sacral). There were very few (<1%) bifurcating (ascending/descending) premotor neurons in the thoracic cord after injections in homolateral GS and FMs (4/523 double-labelled premotor neurons between T2 and T11, *n* = 3, *Figure 6—figure supplement 2A, B*).

We identified premotor long ascending propriospinal neurons (LAPNs) in the lumbar cord, about half of which were localized in the dorsal ipsilateral quadrant (56/117, *n* = 6 forelimb–hindlimb injections). This distribution of lumbar premotor LAPNs is different from that of cervical premotor LDPNs, which were almost exclusively ventral (164/172, *n* = 13 pair of injections, see above). Of the 117 lumbar premotor LAPNs identified, 10 were also labelled from GS injections, indicating that some neurons projected both to local lumbar MNs as well as to cervical MNs (*n* = 6 ipsilateral forelimb–hindlimb injections, *Figure 6—figure supplement 2C, D*). However, the position of these particular divergent premotor LAPNs was different from that of the premotor LDPNs, in that they were not localized within one quadrant of the cord (*Figure 6—figure supplement 2D*).

Finally, we turned our attention to the sacral spinal cord, where we found few premotor LAPNs (12 neurons in four of six mice). Of these, however, 10/12 were in the ventral contralateral quadrant (*n* = 6 ipsilateral forelimb–hindlimb injections, *Figure 6—figure supplement 2E, F*), similar to the location of the cervical premotor LDPNs. Like these cervical neurons, the sacral LAPNs had strikingly large somata

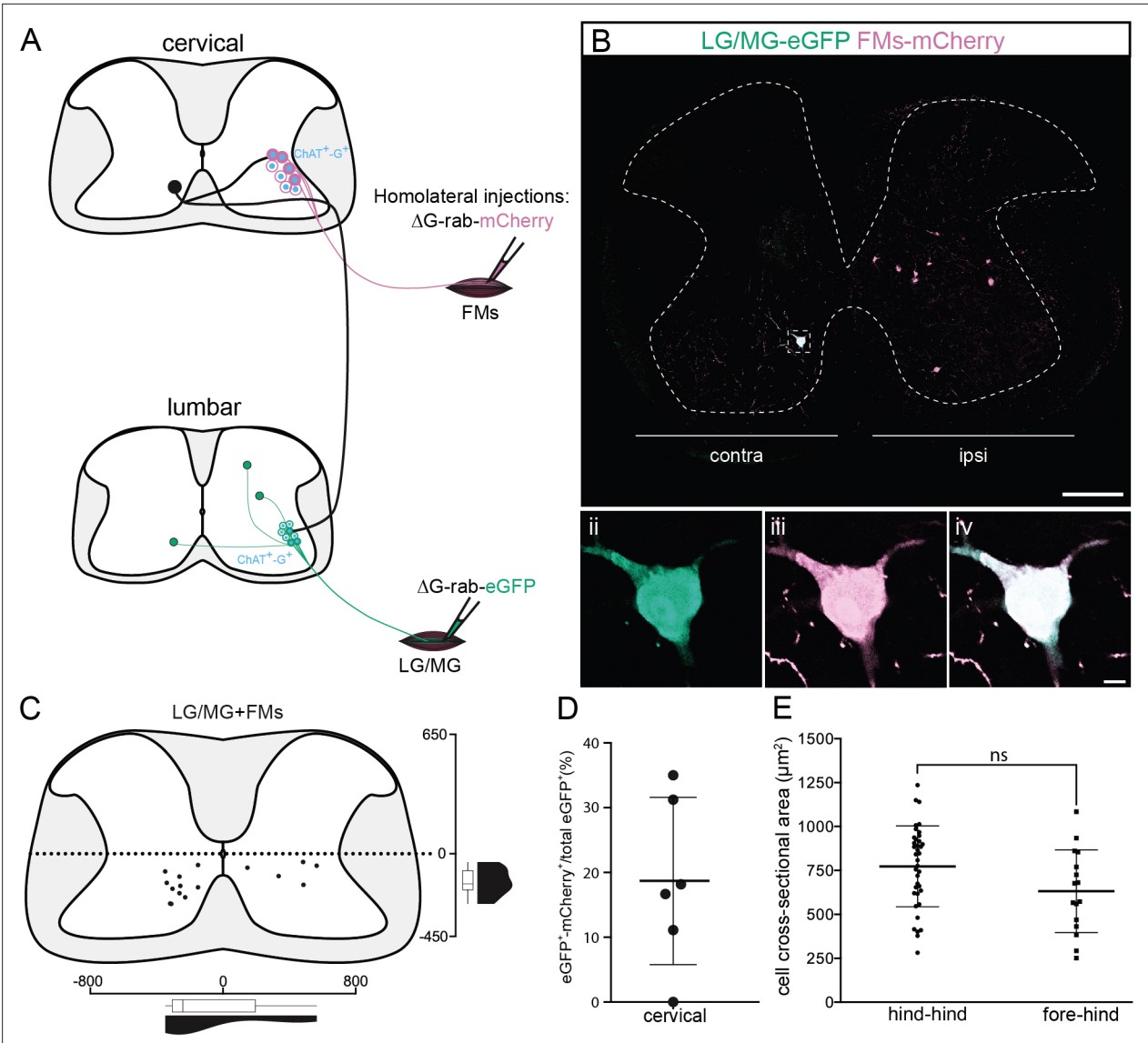

**Figure 6.** Cervical premotor long descending propriospinal neurons (LDPNs) innervate homolateral lumbar and cervical motoneurons (MNs). (**A**) Experimental strategy to determine whether divergent cervical premotor LDPNs that innervate homolateral lumbar and cervical MNs do exist. (**Bi**) Representative example of a transverse section from the cervical cord following an injection in forearm muscles (FMs) (ΔG-Rab-mCherry) and GS (ΔG-Rab-eGFP) muscles, showing a premotor LDPN infected from the two contralateral motor pools. The dashed box highlights the divergent premotor LDPN. The dashed line drawn outlines the grey matter contour. Dashed box area at higher magnification, showing (**ii**) eGFP, (**iii**) mCherry, and (**iv**) the overlay. (**C**) Distribution of the premotor LDPNs infected from the homolateral injections in GS and FMs. The violin and box plots show the distribution of divergent premotor LDPNs innervating homolateral local FMs and distant GS motor pools along the medio-lateral and dorso-ventral axis. Each violin area is normalized to 1. (**D**) Proportion of cervical premotor LDPNs that also project to FM motor pools per animal. (**E**) Plot showing the sectional area of the cervical divergent premotor LDPNs that diverge to two pools of lumbar MNs (hind_hind) and to the pools of GS and FM MNs (fore_hind). When not specified numbers along the axis indicate distances (in μm). Scale bars: (Bi) 200 μm; (Biv) 10 μm.

The online version of this article includes the following figure supplement(s) for figure 6:

**Source data 1.** Source data for *Figure 6C–E*.

**Figure supplement 1.** Cervical premotor long descending propriospinal neurons (LDPNs) innervate ipsilateral cervical and contralateral lumbar motoneurons (MNs).

**Figure supplement 2.** Premotor long ascending propriospinal neurons (LAPNs) are distributed in the thoracic, lumbar, and sacral spinal cord, and diverge to homolateral lumbar and cervical motoneurons (MNs).

**Figure supplement 2—source data 1.** Source data for *Figure 6—figure supplement 2B, D, F, G*.

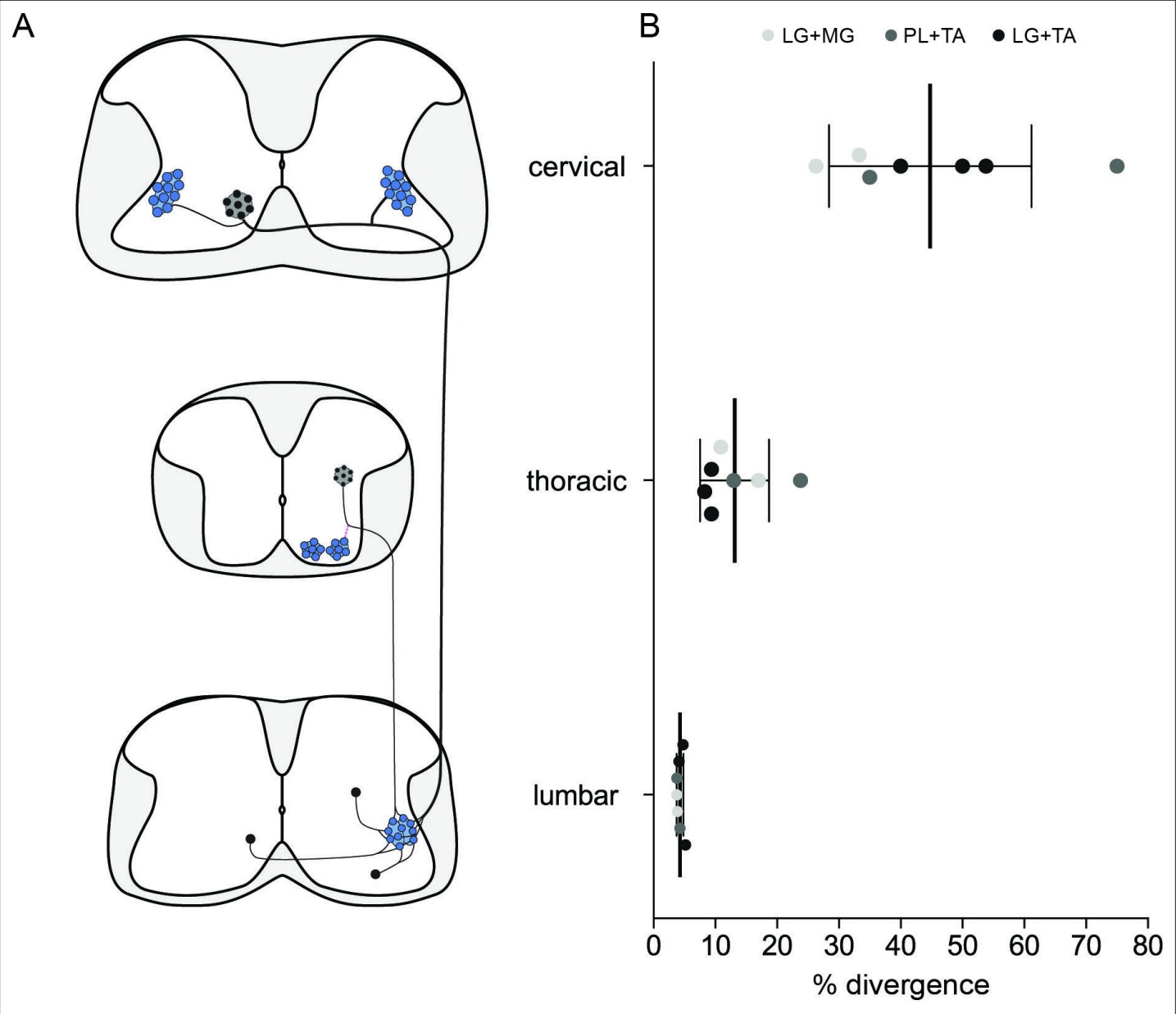

**Figure 7.** Divergence rates throughout the spinal cord and circuits. (**A**) Schematic summarizing the projections determined. (**B**) Plot showing the increase of the apparent divergence rate with the distance between innervated motoneurons (MNs) and premotor neurons.

The online version of this article includes the following figure supplement(s) for figure 7:

**Source data 1.** Source data for *Figure 7B*.

(710 ± 310 μm², *n* = 11 premotor LAPNs, *Figure 6—figure supplement 2G*). Of 12 labelled neurons, 3 were also infected from the hindlimb (LG) injections (*Figure 6—figure supplement 2E–G*). Given that the size and location of these sacral premotor LAPNs were similar to the population of cervical divergent premotor LDPNs, they may represent a 'reverse counterpart' of this descending system.

## Discussion

Animals perform rich repertoires of movements through controlling muscle contractions around joints to produce the fundamental syllables of movement (*Brownstone, 2020*). To understand how behavioural repertoires are formed, it is important to understand the organization of the neural

circuits underlying the production of each syllable. By using monosynaptic restricted RabV tracing techniques, we investigated the presence of spinal premotor INs that project to multiple motor pools and could thus potentially comprise circuits underlying co-activation (joint stiffening) or co-inhibition (joint relaxation) of motor pools across joints and between limbs. We found that at least 1/25 local lumbar premotor INs projects to multiple motor pools, in similar proportions whether these pools were synergist or antagonist pairs. Furthermore, we found that whereas the density of premotor neurons decreases with distance rostral to the motor pool targeted, a high proportion of labelled cervical LDPNs projects to multiple motor pools. These premotor LDPNs are in contralateral lamina VIII, have large somata, are neither glycinergic nor cholinergic, and project to multiple motor pools including those in the lumbar and cervical enlargements. These divergent neurons could thus form a substrate for joint and multi-joint stiffening that contributes to the production of a fundamental syllable of movement.

## Estimating proportions of divergent premotor INs

The control of MNs across motor pools through spinal premotor circuits is required for the performance of all motor tasks involving limb movements. Previous studies showed the importance of motor synergies in the production of complex movements (*Giszter, 2015*; *Takei et al., 2017*), with the spinal cord identified as a potential site for muscle synergy organization (*Bizzi and Cheung, 2013*; *Levine et al., 2014*). In this regard, it might be expected that a significant proportion of local spinal premotor INs innervate multiple motor pools, in particular those corresponding to synergist muscles. Perhaps surprisingly, we found similar rate of divergence throughout the spinal cord be the targeted MN pools synergist or antagonist. In the lumbar region, at least 4 % of the local premotor INs project to two motor pools. More remotely, in thoracic as well as cervical premotor circuits, the apparent rate of divergence was higher but with a decreased density of labelled premotor neurons. Regardless of the proportion of divergent premotor neurons amongst the total premotor population, it is possible that these neurons effectively modulate the synchrony of MN activation and participate in co-activation or co-inhibition of different MN populations.

What proportion of premotor neurons project to more than one motor pool? To investigate the presence of premotor neurons projecting to multiple motor pools in the spinal cord, we used RabV tracing, injecting ΔG-RabV expressing eGFP or mCherry into different pairs of muscles. Although this technique allowed for visualization of divergent premotor neurons throughout the spinal cord, the proportion of divergent premotor neurons has undoubtedly been underestimated. A divergent neuron will be double labelled only if each virus has been efficiently transmitted across its synapses with MNs from both motor pools. Therefore, due to the stochastic nature of the process of crossing a synapse, any given transfer efficiency lower than 100 % will inevitably give rise to an underestimate of the real number of divergent neurons. The efficiency of trans-synaptic jumps for the SADB19 RabV that we used is unknown, and may depend in part on the type of synapse, with stronger connections facilitating transmission of the virus (*Ugolini, 2011*). The only indirect indication of efficiency comes from the direct comparison of the SADB19 and the more efficient CVS-N2c strains, for which there was at least a fourfold increase in the ratio of local secondary to primary infected premotor INs (*Reardon et al., 2016*). This result suggests that the trans-synaptic efficiency of SADB19 is no higher than 25 %. While there is no evidence for a bias towards stronger or weaker synapses (i.e., the actual number of physical contacts) between proximal and distal premotor INs, such a bias could affect efficiency of viral transmission, and could thus also have potentially skewed our relative estimate of divergence. With the simplifying assumption that the efficiencies of viral transfer are equal and independent from each other across spinal cord regions, we simulated a double injection experiment, extracting a binomial distribution, and calculated the relation between the observed and true rate of divergence. With a jump efficiency of 25%, the 4 % divergence rate we observed in the lumbar spinal cord would correspond to an actual rate of divergence of 18 % (*Figure 8*). And this calculated rate is almost certainly an underestimate because of the phenomenon of viral interference, whereby there is a reduced probability of subsequent infection with a second RabV after a window of a few hours after the first infection (*Ohara et al., 2009*). It is therefore likely that the actual rate of divergence of premotor circuit throughout the cord is substantially higher than we observed. Specifically, it is possible that the vast majority of, if not all, premotor LDPNs innervate more than one motor pool.

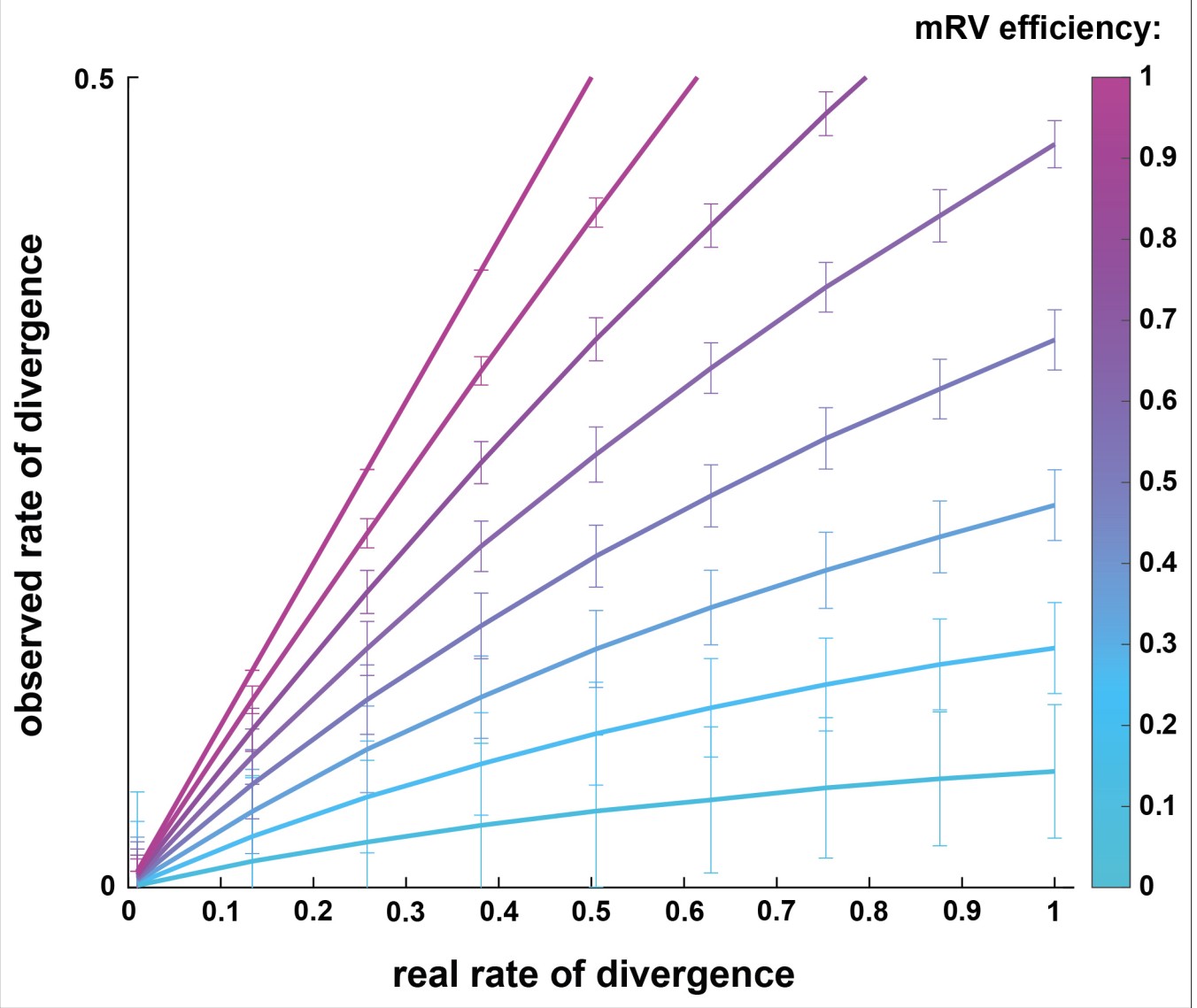

**Figure 8.** Simulation comparing observed vs real rates of divergence depending on trans-synaptic mRV efficiency. Simulation of the spreading of mRV in premotor circuits following double injections, extracted from a binomial distribution. Plot showing the relation between observed rate of divergence depending on the real rate of divergence within premotor spinal circuits. This simulation was run with the simplifying assumption that the efficiencies of viral transfer are equal and independent from each other across spinal cord regions.

## Mapping premotor circuits using the ChAT-Cre;RΦGT mouse

In our experimental model, the rabies glycoprotein is expressed only in neurons expressing ChAT, such as MNs. By restricting primary infection to specific MNs via intramuscular injection of RabV, trans-synaptic viral spread was thus restricted to neurons presynaptic to the infected MN population. It is therefore theoretically possible that there might be double jumps via other presynaptic cholinergic neurons such as medial partition neurons $V0_C$ neurons (**_Zagoraiou et al., 2009_**). MNs also form synapses with other MNs (**_Bhumbra and Beato, 2018_**), so it could also be possible that specificity is lost due to second-order jumps via these cells. We consider double jumps unlikely for two main reasons: (1) following muscle injections, the first trans-synaptic labelling occurs after 5–6 days. Since the tissue was fixed 9 days after injections, it is unlikely that many secondary jumps could have occurred in such a brief time window. And (2) most presynaptic partners of $V0_C$ INs are located in the superficial dorsal laminae (**_Zampieri et al., 2014_**), a region in which we did not observe any labelled INs. We are thus confident that the labelled neurons are premotor. We also acknowledge the possibility that some of the labelled premotor cells might originate from tertiary infection originating from

secondary infection of synaptically connected MNs (*Bhumbra and Beato, 2018*). Such events might be rare (*Ronzano et al., 2021*) and would not alter our findings on the organization of divergent premotor neurons, since we have shown that their distributions are similar, regardless of the particular pair of injected muscles.

## Premotor INs innervating antagonist motor pools: implications for movement

The similar rate of divergence between synergist and antagonist pairs might be surprising. But divergence to agonist and antagonist motor pools has been shown in adult mice (*Gu et al., 2017*), indicating that these circuits are not limited to an early developmental stage. Apart from the cervical divergent premotor LDPNs that are likely to represent a rather homogenous group of excitatory neurons, the divergent premotor neurons in the thoracic and lumbar regions could be comprised of different neural populations, with a mixed population of excitatory, inhibitory, and, in lower proportion, cholinergic neurons (*Figure 1—figure supplement 6* and *Figure 4—figure supplement 1*). These INs that project to antagonist motor pools could thus be involved in modulating either joint stiffening (excitatory) or relaxation (inhibitory). For example, during postural adjustment and skilled movements, divergent excitatory premotor INs would lead to co-contraction of antagonist muscles to facilitate an increase in joint stiffness and to promote stability (*Hansen et al., 2002*; *Nielsen and Kagamihara, 1993*; *Nielsen and Kagamihara, 1992*). In invertebrates, co-contraction of antagonist muscles has also been described preceding jumping (*Pearson and Robertson, 1981*): co-contraction could thus also be important for the initiation of movement.

On the other hand, divergent inhibitory premotor neurons would lead to joint relaxation. This phenomenon is less well studied (*Leis et al., 2000*; *Manconi et al., 1998*). One example could be their involvement in the loss of muscle tone that accompanies rapid eye movement sleep (*Uchida et al., 2021*; *Valencia Garcia et al., 2018*).

## Projections of LDPNs to multiple motor pools

In the cat, long descending fibres originating in the cervical cord have been shown to innervate lumbar MNs (*Giovanelli Barilari and Kuypers, 1969*) and trigger monosynaptic potentials (*Jankowska et al., 1974*). The existence of LDPNs has been confirmed anatomically in neonatal mice (*Ni et al., 2014*) and functionally in adult cats (*Alstermark et al., 1987a*; *Alstermark et al., 1987b*), where they are thought to play a role in posture and stability. Our study confirms the existence of premotor LDPNs, and also indicates that they have a high rate of divergence (up to ~40 % compared to ~13 % for thoracic neurons). Most cervical LDPNs are clustered in contralateral lamina VIII, are virtually all excitatory, and have a distinct morphology with somal size ~ twofold larger than other local cells (and similar to MNs). These findings contrast with the divergent premotor neurons found in the thoracic spinal cord: these are distributed in ipsilateral lamina VI and VII as well as in contralateral lamina VIII and thus clearly comprise multiple neuronal populations. In contrast to thoracic divergent premotor neurons, cervical LDPNs may thus have a more unifying function. Given their apparent widespread divergence, it is possible that these LDPNs are involved in producing widespread increases in muscle tone.

One step towards being able to further assess the function of this population of INs would be through understanding their lineage. Given the poor detection of the Lhx1 transcription factor in postnatal mice (*Figure 5—figure supplement 1*), we could not conclude that the labelled cervical LDPNs are a population that derive from the V0$_v$ or dI2 class. Although in the chick, dI2 INs do not project to MNs (*Haimson et al., 2021*), it is possible that they could in the mouse: these are large neurons located in the ventromedial spinal cord (*Haimson et al., 2021*), and express Lhx1 (*Avraham et al., 2009*). Further experiments using a Dbx1-IRES-GFP mouse line (*Bouvier et al., 2010*), for example, could help to determine the identity of these divergent cervical LDPNs. Genetic access to this particular set of INs would also allow the design of experiments aimed at acute and specific activation or inactivation of divergent LDPNs, and could unravel their anatomy and function in behaviour.

## Concluding remarks

The completion of movements requires well-controlled muscle contractions across multiple joints within and between limbs. The control of any one joint is analogous to the production of syllables

of speech, with the three most fundamental syllables of movement being a change in joint angle (requiring reciprocal inhibition of flexors and extensor MNs), a stiffening of a joint (requiring co-activation of flexors and extensor MNs), and a relaxation of a joint (requiring co-inhibition of flexor and extensor MNs). While neural circuits for reciprocal inhibition have been well studied over many decades (*Eccles, 1969*; *Eccles et al., 1956*), circuits for stiffening or relaxation have not been. Our anatomical data identify neurons that could be potentially implicated in these circuits and show that they are present within and distributed throughout the spinal cord. Thus, the mechanisms that lead to the production of the fundamental syllables of movement could be contained within the spinal cord itself.

# Materials and methods

## Key resources table

| Reagent type (species) or resource | Designation | Source or reference | Identifiers | Additional information |
|---|---|---|---|---|
| Strain, strain background (Rabies virus) | ΔG-Rab-eGFP | Gift from M. Tripodi lab, LMCB Cambridge | | G-deleted Rabies virus |
| Strain, strain background (Rabies virus) | ΔG-Rab-mCherry | Gift from M. Tripodi lab, LMCB Cambridge | | G-deleted Rabies virus |
| Strain, strain background (*Mus musculus*) | ChAT-IRES-Cre | Jackson Laboratory | IMSR Cat# JAX:006410; RRID:IMSR_JAX:006410 | Allele symbol: Chat$^{tm2(cre)Lowl}$; maintained on a C57BL6/J background |
| Strain, strain background (*Mus musculus*) | RΦGT | Jackson Laboratory | IMSR Cat# JAX:024708; RRID:IMSR_JAX:024708 | Allele symbol: Gt(ROSA)26 Sortm1 (CAG-RABVgp4,-TVA)Arenk; maintained on a C57BL6/J background |
| Strain, strain background (*Mus musculus*) | GlyT2-eGFP | Gift from H. Zeilhofer lab, University of Zurich | IMSR Cat# RBRC04708; RRID:IMSR_RBRC04708 | Allele symbol: Tg(Slc6a5-EGFP) 1Uze; maintained on a C57BL6/J background |
| Cell line (*Homo sapiens*, female) | HEK293t/17 | Gift from M. Tripodi lab, LMCB Cambridge | RRID:CVCL_1926 | ATCC, cat. no. CRL-1126 |
| Cell line (*Mesocricetus auratus*, male) | BHK-21 | Gift from M. Tripodi lab, LMCB Cambridge | RRID: CVCL_1915 | ATCC # CCL-10 |
| Cell line (*Mesocricetus auratus*, male) | BHK-G | Gift from M. Tripodi lab, LMCB Cambridge | RRID: CVCL_1915 | Modified from ATCC Cat# CCL-10; RRID: CVCL_1915 to express the rabies glycoprotein |
| Antibody | anti-ChAT (Goat polyclonal) | Millipore | Cat# AB144P; RRID:AB_2079751 | IF (1:100) |
| Antibody | anti-mCherry (Chicken polyclonal) | Abcam | Cat# ab205402; RRID:AB_2722769 | IF (1:2500) |
| Antibody | anti-GFP (Rabbit polyclonal) | Abcam | Cat# ab290; RRID:AB_303395 | IF (1:2500) |
| Antibody | anti-vGluT2 (Guinea pig polyclonal) | Millipore | Cat# AB2251-I; RRID:AB_2665454 | IF (1:2500) |
| Antibody | anti-Lhx1 (Rabbit polyclonal) | Gift from T. Jessell lab, Columbia University, New York | | IF (1:5000) |
| Antibody | anti-Rabbit IgG H&L Alexa Fluor 647 (Donkey polyclonal) | Abcam | Cat# ab150079; RRID:AB_2722623 | IF (1:1000) |

*Continued on next page*

*Continued*

| Reagent type (species) or resource | Designation | Source or reference | Identifiers | Additional information |
|---|---|---|---|---|
| Antibody | anti-Goat IgG H&L Alexa Fluor 405 (Donkey polyclonal preadsorbed) | Abcam | Abcam Cat# AB175665; RRID:AB_2636888 | IF (1:200) |
| Antibody | anti-Rabbit IgG H&L Alexa Fluor488 (Donkey polyclonal Highly Cross-Adsorbed) | Thermo Fisher Scientific | Cat# A-21206; RRID:AB_2535792 | IF (1:1000) |
| Antibody | anti-Chicken IgY (IgG) H&L Cy3-AffiniPure (Donkey polyclonal) | Jackson Immuno Research Labs | Cat# 703-165-155; RRID:AB_2340363 | IF (1:1000) |
| Chemical compound, drug | Mowiol 4–88 | Sigma-Aldrich | Cat# 81381–250 G | |
| Software, algorithm | ZEN Digital Imaging for Light Microscopy: Zen Blue 2.3 | Carl Zeiss light microscopy imaging systems | RRID:SCR_013672 | |
| Software, algorithm | Imaris 9.1 | Bitplane | RRID:SCR_007370 | |
| software, algorithm | R 3.6.2 | R Project for Statistical Computing | RRID:SCR_001905 | |
| Software, algorithm | Prism 7.0 | GraphPad | RRID:SCR_002798 | |
| Software, algorithm | Adobe illustrator version CC 2019 | Adobe | RRID:SCR_010279 | |

## Mouse strains

All experiments (*n* = 27) were performed according to the Animals (Scientific Procedures) Act UK (1986) and certified by the UCL AWERB committee, under project licence number 70/7621. Homozygous ChAT-IRES-Cre mice (which have an IRES-Cre sequence downstream of the ChAT stop codon, such that Cre expression is controlled by the endogenous ChAT gene promoter without affecting ChAT expression; *Rossi et al., 2011*, Jackson lab, stock #006410) crossed with homozygous RΦGT mice (*Takatoh et al., 2013*, Jackson lab, stock #024708), that have Cre dependent expression of the rabies glycoprotein and the avian viral receptor TVA, whose expression is not employed in this study were used for double injections (see the detail of animal use for each type of injection). For single injections, homozygous ChAT-IRES-Cre mice (termed ChAT-Cre here) were crossed with hemizygous GlyT2-eGFP mice (BAC transgene insertion in exon 2 of *Slc6a5* gene allowing specific eGFP expression in GlyT2-positive cells, MGI:3835459, *Zeilhofer et al., 2005*) and their eGFP-positive offspring was mated with homozygous RΦGT (see *Supplementary file 3*).

## Virus production, collection, and titration

We used the glycoprotein G-deleted variant of the SAD-B19 vaccine strain rabies virus (a kind gift from Dr M. Tripodi). Modified RabV (ΔG-Rab) with the glycoprotein G sequence replaced by mCherry or eGFP (ΔG-Rab-eGFP/mCherry) was produced at a high concentration with minor modifications to the original protocol (*Osakada et al., 2011*). BHK cells expressing the rabies glycoprotein G (BHK-G cells) were plated in standard Dulbecco modified medium with 10 % foetal bovine serum (FBS) and split after 6–7 hr incubating at 37°C and 5 % $CO_2$. They were inoculated at a multiplicity of infection of 0.2–0.3 with either ΔG-Rab-eGFP or mCherry virus in 2 % FBS, and incubated at 35°C and 3 % $CO_2$. Plates were then split in 10 % FBS at 37°C and 5 % $CO_2$. After 24 hr the medium was replaced by 2 % FBS medium and incubated at 35°C and 3 % $CO_2$ for 3 days (virus production). The supernatant was collected and medium was added for another cycle (three cycles maximum), after which the supernatant was filtered (0.45 μm filter) and centrifuged 2 hr at 19,400 rpm (SW28 Beckman rotor). The pellets were re-suspended in phosphate-buffered saline (PBS) and centrifuged together at 21,000 rpm, 4°C, 4 hr in a 20 % sucrose gradient. Pellets of each collection were then re-suspended and stored in 5–10 μl aliquots at –80°C.

Virus titration was performed on BHK cells plated in 10 % FBS medium at $1.5 \times 10^5$ cells/ml and incubated overnight at 37°C and 10 % $CO_2$ (growth). The virus was prepared for two serial dilutions with two different aliquots and added in the well after an equal volume of medium had been removed (serial dilution from $10^{-3}$ to $10^{-10}$) and incubated 48 hr at 35°C and 3 % $CO_2$. The titre was determined from the count of cells in the higher dilution well and was between $10^9$ and $10^{10}$ infectious units (IU)/ml.

## Intramuscular injections

A subcutaneous injection of analgesic (carprofen, 1 µl, 10% wt/vol) was given to the neonatal pups (P1–P3) prior to surgery and all procedures were carried out under general isoflurane anaesthesia. After a skin incision to expose the targeted muscle, the virus (1 µl) was injected intramuscularly using a Hamilton injector (model 7652-01) mounted with a bevelled glass pipette (inner diameter 50–70 µm). The mice were injected in TA and PL (ankle flexor pair), LG and MG (ankle extensor pair) for synergist pairs and TA and LG for antagonist pairs. In hindlimb/forelimb double injections, the LG and MG were both injected with 1 µl of one RabV to increase the number of long projecting cells infected. In addition, 1 µl of the second RabV was injected in FMs (see *Supplementary file 2*) without selecting a specific muscle. The injected viruses were used at a titre between $10^9$ and $10^{10}$ IU/ml. The incisions were closed with vicryl suture, and the mice were closely monitored for 24 hr post-surgery. Mice were perfused 9 days after the injections. Due to the proximity of synergist pairs of muscles, prior to spinal tissue processing, we dissected the injected leg and confirmed that there was no contamination of virus across the injected muscles or in adjacent muscles below or above the knee. When injecting FMs, we could not target a single muscle. To visualize which muscles had been infected, we carefully dissected each FM and assess for the presence of fluorescent signal (see *Supplementary file 2*). Three heterozygous RΦGT mice were also injected (LG muscle) with an EnvA pseudotyped RabV in order to test simultaneously for ectopic expression of G or of the TVA receptors. In three control animals we observed one to three labelled MNs, but no IN labelling. This indicates the presence of minimal ectopic TVA expression, but not of G (*Ronzano et al., 2021*).

## Tissue collection and immunohistochemistry

The mice were perfused with PBS (0.1 M) followed by PBS 4 % paraformaldehyde under terminal ketamine/xylazine anaesthesia (i.p. 80 and 10 mg/kg, respectively). The spinal cords were then collected through a ventral laminectomy and post-fixed for 2 hr. The cords were divided into the different parts of the spinal cord (cervical [C1–T1], thoracic [T2–T11], lumbar [L1–L6], and sacral [S1–S4]), cryoprotected overnight in 30 % sucrose PBS, embedded in optimal cutting temperature compound (Tissue-Tek) and sliced transversally (30 µm thickness) with a cryostat (Bright Instruments, UK). Sections were incubated with primary antibodies for 36 hr at 4°C and with secondary antibodies overnight at 4°C in PBS double salt, 0.2 % Triton 100 -X (Sigma), 7 % donkey normal serum (Sigma). The primary antibodies used were: goat anti-choline acetyltransferase (ChAT, 1:100, Millipore, AB144P), chicken anti-mCherry (1:2500, Abcam, Ab205402), rabbit anti-GFP (1:2500, Abcam, Ab290), guinea pig anti-vGluT2 (1:2500, Millipore, AB2251-I), and rabbit anti-Lhx1 (1:5000, from Dr. T Jessell, Columbia University, New York); and the secondary antibodies: donkey anti-rabbit Alexa 647 (1:1000, Abcam, Ab150079), donkey anti-goat preadsorbed Alexa 405 (1:200, Abcam, Ab175665), donkey anti-rabbit Alexa 488 (1:1000, Thermo Fisher, A21206), and donkey anti-chicken Cy3 (1:1000, Jackson ImmunoResearch, #703-165-155). The slides were mounted in Mowiol (Sigma, 81381-250 G) and coverslipped (VWR, #631-0147) for imaging.

## Confocal imaging and analysis

Images of the entire sections were obtained using a Zeiss LSM800 confocal microscope with a ×20 air objective (0.8 NA) and tile advanced set up function (ZEN Blue 2.3 software). A ×63 oil objective was used for Airy scan imaging of somata and excitatory boutons. Tiles were stitched using Zen Blue and analyses were performed using Zen Blue and Imaris (Bitplane, version 9.1) software packages. Location maps were plotted setting the central canal as (0,0) in the ($x$,$y$) Cartesian system and using the 'Spots' function of Imaris. The $y$-axis was set to the dorso-ventral axis. Positive values were assigned for dorsal neurons in the $y$-axis and ipsilateral (to the hindlimb injection) neurons in the $x$-axis. Coordinates were collected on every section and normalized through the cervical, thoracic, lumbar, and sacral parts separately using grey matter borders and fixing the width and the height of the transverse

hemisections. To calculate divergence rates, given the high density of premotor INs infected in the lumbar cord all infected premotor INs (eGFP+, mCherry+ and eGFP+ mCherry+) were quantified in one of every three sections which further allowed to avoid counting the same cells twice on consecutive sections. In the cervical, thoracic, and sacral regions, all cells were quantified, as their low density allowed for manually excluding premotor neurons found in consecutive sections. Since MNs are big cells localized as a restricted column of the ventral spinal cord, we quantified them on every other sections, to avoid counting the same cell twice on consecutive sections.

## Statistics

All statistical analyses and plots were made using R (R Foundation for Statistical Computing, Vienna, Austria, 2005, http://www.r-project.org, version 3.6.2) and GraphPad PRISM (version 7.0). To compare cell sectional areas, non-parametric rank tests were used as specified in each related result. The numbers of animals/cells in each experiment and statistical tests used are reported in the figure legends or directly in the text. Results and graphs illustrate the mean ± standard deviation. Statistical significance levels are represented as follows: $*p < 0.05$, $**p < 0.01$, $***p < 0.001$, $****p < 0.0001$, and ns: not significant.

## Acknowledgements

We are grateful to Dr N Zampieri and all the members of Brownstone and Beato labs for insightful comments on the manuscript. We want to thank S Morton for providing Lhx1 antibodies, Prof L Greensmith for access to her lab facilities, and Dr MG Özyurt for helping with immunostaining. This work was supported by a Leverhulme Trust grant (grant number RPG-2013-176) and a BBSRC grant (BB/L001454) to MB and a Wellcome Trust Investigator Award to RMB (110193). RMB is supported by Brain Research UK.

## Additional information

### Funding

| Funder | Grant reference number | Author |
| --- | --- | --- |
| Leverhulme Trust | RPG-2013-176 | Marco Beato |
| Biotechnology and Biological Sciences Research Council | BB/L001454 | Marco Beato |
| Wellcome Trust | 110193 | Robert M Brownstone |
| Brain Research UK | | Robert M Brownstone |

The funders had no role in study design, data collection and interpretation, or the decision to submit the work for publication.

### Author contributions

Remi Ronzano, Conceptualization, Data curation, Formal analysis, Investigation, Methodology, Software, Validation, Visualization, Writing – original draft, Writing – review and editing; Camille Lancelin, Conceptualization, Data curation, Formal analysis, Investigation, Methodology, Supervision, Visualization, Writing – review and editing; Gardave Singh Bhumbra, Conceptualization, Data curation, Formal analysis, Investigation, Methodology, Software, Supervision, Writing – review and editing; Robert M Brownstone, Conceptualization, Data curation, Funding acquisition, Investigation, Project administration, Resources, Supervision, Writing – original draft, Writing – review and editing; Marco Beato, Conceptualization, Data curation, Formal analysis, Funding acquisition, Investigation, Methodology, Project administration, Resources, Software, Supervision, Writing – original draft, Writing – review and editing

### Author ORCIDs

Remi Ronzano (iD) http://orcid.org/0000-0002-4927-9474

Robert M Brownstone  http://orcid.org/0000-0001-5135-2725
Marco Beato  http://orcid.org/0000-0002-7283-8318

## Ethics

All experiments were performed in strict adherence to the Animals (Scientific Procedures) Act UK (1986) and certified by the UCL AWERB committee, under project licence number 70/7621. All surgeries were performed under general isofluorane anaesthesia and before surgery animals were injected subcutaneously with an analgesic (carprofen, 1 µl, 10% wt/vol) and the mice were closely monitored for a 24-hr period following surgery to detect any sign of distress or motor impairment. Every effort was made to minimize suffering.

## Decision letter and Author response

Decision letter https://doi.org/10.7554/eLife.70858.sa1
Author response https://doi.org/10.7554/eLife.70858.sa2

---

# Additional files

## Supplementary files

• Transparent reporting form

• Supplementary file 1. Numbers of MNs and premotor neurons, and medio-lateral and dorso-ventral distributions of divergent premotor neurons across individual experiments. Distribution of divergent premotor neurons per region of the spinal cord across individual experiments, expressed as median± first/third quartile.

• Supplementary file 2. Details of muscles infected following forearm injections. (+) means that a fluorescent signal was found in muscle fibres, (−) means that no fluorescence was observed following muscle dissections.

• Supplementary file 3. List of the mice used for each experiment, including genotype and figure in which they are shown.

## Data availability

All data generated during this study are included in the manuscript and supporting files. Source data files have been provided for Figures 1 to 7 in the main text and for Figure 1 – figure supplement 4-5, Figure 2 – figure supplement 4, Figure 3 – figure supplement 4, and Figure 6 – figure supplement 2.

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
