## [Editor Report]

This manuscript uses viral tracing to identify interneurons, throughout the spinal cord, which synapse onto motoneurons innervating pairs of flexor and extensor hindlimb muscles. Importantly, the data identifies single premotor interneurons which travel to, and presumably regulate the activity of, multiple motor pools. It is possible that these premotor neurons are involved in regulating muscle stiffness across a joint.

---

## [Decision Letter]

**Decision letter after peer review:**

Thank you for submitting your article "Spinal neurons innervating multiple local and distant motor pools" for consideration by *eLife*. Your article has been reviewed by 3 peer reviewers, and the evaluation has been overseen by a Reviewing Editor and Ronald Calabrese as the Senior Editor. The following individual involved in review of your submission has agreed to reveal their identity: Simon Gosgnach (Reviewer #1).

The three reviewers acknowledge that the paper is of a great interest for researchers working within the field of motor control, and more specifically at the level of the spinal cord. Despite the rather descriptive aspect of the study it might indeed set the foundation for future studies regarding the specific identities and functions of the premotor neurons investigated here. Their detailed comments are described below, but I would like to emphasize a few important points harvested from their reviews:

1) You should tone down the part on the V0v origin identification, or better do the appropriate characterization as suggested below;

2) You should further discuss the neurotransmitter characterization and eventually address this point experimentally if possible;

3) You should further describe the rostro-caudal extension of the labelled pre-motoneurons.

4) Also, a functional aspect is clearly missing, the more you can address this in your revised version the more it will strengthen the paper, but we are aware that fully investigating this would require too much time and represents a significant amount of work if the data are not available yet.

*Reviewer #1 (Recommendations for the authors):*

1. More detail regarding rostra-caudal location in the spinal cord should be given in all cases. First I think a Figure describing the location of the motor pools for the 4 injected muscles should be included- this could be in the supplementary data or it could go as a panel in Figure 1. This is needed so we have a clear idea of how far the axons of the PMIs project. In my experience a motor pool can span several lumbar segments. More specific information should also be provided regarding the location of the sections which contain the PMIs being described. I realize this is not an exact science after cutting tissue, but given your care in counting every third section, you should have a pretty good idea of which segment houses the neurons. This would be preferred over the current description of "rostral cervical" or equivalent.

In the Figures the number of MNs infected with either virus is typically quite low so I assume the sections are not particularly close to the middle of the infected motor pools?

2. The amount of raw data shown is light. There is one immuno panel shown per Figure. I realize the take home message comes from the schematics, but it would be nice to see images of spinal sections with more single and double infected cells. Each immuno image contains just one divergent PMI and other than Figure 1 and 6 there are no other cells that seem to be infected with either the red or green expressing virus in the immuno image.

3. I get a little uncomfortable with the data interpretation at the back end of the Results- particularly with the section titled Cervical LDPNs are excitatory- this is speculation and should not be in the title. The data simply shows 95% of these cells are not glycinergic.

4. Related to point 3 – The next section is titled "cervical premotor LDPNs likely arise from the V0v domain". In addition to V0 and V1 cells, Lhx1 is also expressed postmitotically in the dI2, dI4, dI6, and V2 populations, and even some V3 cells (Alanyick et al., 2012, Delile….Sagner biorxiv paper https://doi.org/10.1101/472415). It is thus extremely speculative to suggest that the V0v cells are the ones we are looking at here. Even more so when we factor in the finding that Lhx1 plays a key role in establishing GABAergic neuronal identity (Pillai et al., 2007).

Incidentally the current thinking on V0v cells is not that they contact MNs directly, but rather disynaptically via an inhibitory IN (see Kiehn 2016 Nat Rev Nsci review). Some V0 and V3 neurons have been shown to project to MNs (as is stated in line 200- shown in Lanuza et al., 2004 and Zhang et al., 2008), but V3 cells also have been shown to project to non MN targets recent (Chopek Cell Rep paper), and V0 cells very possibly have many other downstream targets, this simply has yet to be investigated.

I really think this part of the manuscript should not be included due to its preliminary nature. In order to be included there should be some sort of Evx1 staining, or a functional assay in the V0v deficient mice used in the Talpalar 2012 paper in which issues with muscle stiffness are seen in the absence of V0v cells.

The reason I have such an issue with this is that the identity of these cells as belonging to the V0v population is so very circumstantial based on the data here. If it were to be published as is, V0v cells would immediately be cited, in the work of others, as key regulators of muscle stiffness. Even in this manuscript – line 398 of the discussion the authors say that a subset of labeled LDPNs "are clearly V0v derived". This is a problem.

*Reviewer #2 (Recommendations for the authors):*

Could the author present data or mention that they have tested that pure RΦGT mice injected with virus injected do not labelled any premotor neurons?

*Reviewer #3 (Recommendations for the authors):*

– It should be mentioned earlier in the Results that this study was performed in neonates.

– Green and blue in the immuno figures (i.e. sections in Figure 1Bi, Figure 2Ai) are difficult to differentiate between due to low contrast between the two.

– Line 9-11: Divergent is used before it is defined. It works in the initial sentence (line 9) but is a bit deceptive in the next (lines 10-11) as it seems to imply that divergent is referring only to antagonist motor pools.

– Lines 37-42: I suggest adjusting the framing of this section. Most of those questions have answers, as mentioned in the last line of the paragraph and in the Discussion. Instead, highlighting that the location and identities of many of these neurons is unknown may better set up the results to follow.

– Lines 146-148: It is not clear what the 22% refers to. It may be "where they make up 22% of the neurons labelled"?

– The schematic in Figure S4 A implies that the ventral divergent interneuron in cervical cord is a double jump from the forelimb motor pool. It is likely that there is a 'mis-connection' in the schematic – the ventral neuron should synapse with the forelimb motor neurons, rather than the dorsal neuron to the ventral neuron.

---

## [Author Response]

1) You should tone down the part on the V0v origin identification, or better do the appropriate characterization as suggested below;

We have extensively edited the sections related to the identification of V0v interneurons. We agree with the referees that we do not have sufficient evidence to pinpoint the genetic lineage of the long range descending interneurons and have toned this down considerably. As suggested by Reviewer#1 we performed further experiments using an Evx1 antibody to selectively label V0v interneurons. However, while good labeling was obtained in neonatal (P0) tissue, no labelling was detected in tissue of the age we used in the present study (P9-P10), in line with downregulation of transcription factors during development – a feature that also reduced our sensitivity in the Lhx1 assay, albeit to a lesser extent. To compound our uncertainty, a recent study was published while we were revising our manuscript identifying a class of dI2 derived interneurons with morphological features similar to the LDPNs described in our study. And we note that dI2 INs also express Lhx1. Given this further uncertainty and the reduced level of detection of Lhx1 in more mature tissue, we have toned down any conclusion that may suggest we have identified the cardinal class of divergent LDPNs. But we prefer to keep the data in the manuscript since we give evidence that at least some LDPNs are Lhx1-expressing (and thus likely V0v or dI2 as discussed). Specific changes in the text are detailed in the response to reviewers #1 and #3.

2) You should further discuss the neurotransmitter characterization and eventually address this point experimentally if possible;

Identifying the transmitter phenotype of all the divergent interneurons would have required a different protocol, e.g. with a genetically (non-cre) labelled mouse. While the excitatory nature of the divergent LDPNs is sufficiently established, divergent premotor INs in the lumbar and thoracic region are certainly both inhibitory and excitatory. We now demonstrate this point further by adding supplementary figures showing double labelled terminals that are positive or negative for VGlut2 and are localized throughout the thoracic and lumbar cord (Figure 1—figure supplement 6). Further work will be required to understand whether divergent excitatory and inhibitory premotor interneurons are segregated and whether they belong to homogeneous cardinal classes or not.

3) You should further describe the rostro-caudal extension of the labelled pre-motoneurons.

We have added new figures where the distribution of premotor interneurons is shown along the rostrocaudal axis for lumbar, thoracic and sacral cord (see response to reviewer #3 and Figure 1-supplementary figure 4, Figure 2-supplementary figure 4 and Figure 3-supplementary figure 4).

4) Also, a functional aspect is clearly missing, the more you can address this in your revised version the more it will strengthen the paper, but we are aware that fully investigating this would require too much time and represents a significant amount of work if the data are not available yet.

The question on the function of these LDPNs is clearly one of our top priorities. Our manuscript is focused on the discovery of these peculiar cells, but determining their function is a necessary next step. The finding of a precise anatomical location of LDPNs and the availability of tracing methods will now allow us to access and manipulate these cells, but this is clearly a separate project and in our opinion could not be reduced to an appendage to an already dense manuscript.

Reviewer #1 (Recommendations for the authors):1. More detail regarding rostra-caudal location in the spinal cord should be given in all cases. First I think a Figure describing the location of the motor pools for the 4 injected muscles should be included- this could be in the supplementary data or it could go as a panel in Figure 1. This is needed so we have a clear idea of how far the axons of the PMIs project. In my experience a motor pool can span several lumbar segments. More specific information should also be provided regarding the location of the sections which contain the PMIs being described. I realize this is not an exact science after cutting tissue, but given your care in counting every third section, you should have a pretty good idea of which segment houses the neurons. This would be preferred over the current description of "rostral cervical" or equivalent.

We have now added as supplementary figures the rostro-caudal distribution of divergent premotor INs of lumbar segments, and the rostral-caudal distribution of all premotor INs of the thoracic and cervical regions (Figure 1—figure supplement 4, Figure 2—figure supplement 4 and Figure 3—figure supplement 4). We also provided the number of motoneurons infected from each pair of muscles injected, a number that we note is underestimated due to the toxicity of the virus for starter cells (supplementary file 1).

In the Figures the number of MNs infected with either virus is typically quite low so I assume the sections are not particularly close to the middle of the infected motor pools?

In Figure 1, the section presented is close to the middle of the infected motor pools. Indeed, 2 MNs infected on one single section, as in the example of Figure 1, represents the typical count of MNs within a single section within the labelled motor column. This is due partly to the sparseness or rabies infection, but, more importantly, to the fact that each section is only 30 μm thick. We have also added supplementary figures (figure supplement 1,2,3 to Figures 1,2,3) showing more examples of lumbar sections following injections in the different motor pools. Amongst the examples shown, there are up to 8 MNs infected on a single section, but this is a rare occurrence (Figure 1—figure supplement 1A).

2. The amount of raw data shown is light. There is one immuno panel shown per Figure. I realize the take home message comes from the schematics, but it would be nice to see images of spinal sections with more single and double infected cells. Each immuno image contains just one divergent PMI and other than Figure 1 and 6 there are no other cells that seem to be infected with either the red or green expressing virus in the immuno image.

On Figure 1, we highlighted the 3 divergent interneurons on the image shown. Due to the relative low density of divergent premotor neurons and the thinness of the sections, it is impossible to have more divergent neurons in a single section, particularly in the thoracic and cervical cord. To provide a broader perspective of divergent premotor neurons, we added more examples of stained sections from lumbar, thoracic, and cervical spinal cord showing divergent neurons. These are presented in figure supplements 1,2,3 linked to figure 1, 2, and 3.

3. I get a little uncomfortable with the data interpretation at the back end of the Results- particularly with the section titled Cervical LDPNs are excitatory- this is speculation and should not be in the title. The data simply shows 95% of these cells are not glycinergic.

We have modified the title of this section: “Cervical LDPNs are neither inhibitory nor cholinergic” (line 201) since only 1/21 cervical LDPNs was eGFP positive and none of them were ChAT positive.

4. Related to point 3 – The next section is titled "cervical premotor LDPNs likely arise from the V0v domain". In addition to V0 and V1 cells, Lhx1 is also expressed postmitotically in the dI2, dI4, dI6, and V2 populations, and even some V3 cells (Alanyick et al., 2012, Delile….Sagner biorxiv paper https://doi.org/10.1101/472415). It is thus extremely speculative to suggest that the V0v cells are the ones we are looking at here. Even more so when we factor in the finding that Lhx1 plays a key role in establishing GABAergic neuronal identity (Pillai et al., 2007).Incidentally the current thinking on V0v cells is not that they contact MNs directly, but rather disynaptically via an inhibitory IN (see Kiehn 2016 Nat Rev Nsci review). Some V0 and V3 neurons have been shown to project to MNs (as is stated in line 200- shown in Lanuza et al., 2004 and Zhang et al., 2008), but V3 cells also have been shown to project to non MN targets recent (Chopek Cell Rep paper), and V0 cells very possibly have many other downstream targets, this simply has yet to be investigated.I really think this part of the manuscript should not be included due to its preliminary nature. In order to be included there should be some sort of Evx1 staining, or a functional assay in the V0v deficient mice used in the Talpalar 2012 paper in which issues with muscle stiffness are seen in the absence of V0v cells.The reason I have such an issue with this is that the identity of these cells as belonging to the V0v population is so very circumstantial based on the data here. If it were to be published as is, V0v cells would immediately be cited, in the work of others, as key regulators of muscle stiffness. Even in this manuscript – line 398 of the discussion the authors say that a subset of labeled LDPNs "are clearly V0v derived". This is a problem.

We thank the reviewer and agree that our evidence for the identification of V0 interneurons is not definitive. While preparing the revision, we attempted to identify Evx1 derived interneurons using a specific antibody (a gift from Columbia University). However, while we obtained good staining in young (P0) tissue, no staining was observed in P9 tissue, the age at which the tissue was analyzed in this paper. Having tried many different concentrations and incubation times, with similar results, we conclude that Evx1 expression is downregulated, even more than Lhx1 (see Figure 5—figure supplement 1) and therefore, a posteriori identification of the long descending projection neurons is not possible. Only genetic labelling could resolve the issue, but this would not be possible since we do not have the relevant mouse line. To compound our uncertainty, a recent study was published while we were revising our manuscript identifying a class of dI2 derived interneurons with morphological features similar to the LDPNs described in our study. And we note that dI2 INs also express Lhx1. Given this further uncertainty and the reduced level of detection of Lhx1 in more mature tissue, we have toned down any conclusion that may suggest we have identified the cardinal class of divergent LDPNs and re-written the related section, (now starting at line 219). But we prefer to keep the data in the manuscript, since we give evidence that at least some LDPNs are Lhx1-expressing (and thus likely V0v or dI2 as discussed). See also answer to reviewer #2 related to the same point.

Reviewer #2 (Recommendations for the authors):4) Could the author present data or mention that they have tested that pure RΦGT mice injected with virus injected do not labelled any premotor neurons?

We tested for leaks of the rabies glycoprotein by performing injection of an EnvA pseudotyped rabies virus in heterozygous RΦGT mice. We chose to use a pseudotyped rabies in order to test for ectopic expression of both the rabies glycoprotein and the TVA receptor (since these mice contain a floxed cassette expressing both proteins, Takatoh et al., 2013). The example data are shown in our preprint in which we describe a similar approach (Ronzano et al., 2021). In summary, following the injections we observed scant motoneuron labelling (1, 1 and 3 motoneurons in each of the animals tested), but no interneurons. Since we used pseudotyped rabies, the infection of some motoneurons shows that there is some ectopic expression of TVA in motoneurons that gave rise to motoneuron infection due to the extremely high affinity of EnvA for the TVA receptor. However, no trans-synaptic labelling was observed, indicating that any leak of the G protein, was not sufficient to sustain transsynaptic jumps. We now state this at the end of Methods section on intramuscular injections (lines 504-508).

Reviewer #3 (Recommendations for the authors):– It should be mentioned earlier in the Results that this study was performed in neonates.

This has been clarified and is now mentioned line 95 of the results.

– Green and blue in the immuno figures (i.e. sections in Figure 1Bi, Figure 2Ai) are difficult to differentiate between due to low contrast between the two.

Colors are unfortunately difficult to pick when taking into account the many ways people can actually see them. We spend far too much time in lab meetings discussing colours. We had to pick two colors that everyone could easily differentiate for eGFP (green) and mCherry (pink), while keeping colors close to their fluorescent emission spectrum to avoid misunderstanding. Also, since we are focusing on double labelled neurons, the colocalisation of the two colors needed to give a completely different tint (white in our case). Blue remained the only option as the fourth color so that it could readily be differentiated from the three others. It is true that blue is more difficult to visualize, however, since blue is used for motoneuron staining, which is not the focus of the study, we thought that the combination used was the best option.

– Line 9-11: Divergent is used before it is defined. It works in the initial sentence (line 9) but is a bit deceptive in the next (lines 10-11) as it seems to imply that divergent is referring only to antagonist motor pools.

Divergent is now defined clearly in the abstract before its use.

– Lines 37-42: I suggest adjusting the framing of this section. Most of those questions have answers, as mentioned in the last line of the paragraph and in the Discussion. Instead, highlighting that the location and identities of many of these neurons is unknown may better set up the results to follow.

We have included the referee’s suggestion and, while keeping the list of questions, we have stressed that what is not known are the location and identities of the interneurons mediating synergies (lines 46-48).

– Lines 146-148: It is not clear what the 22% refers to. It may be "where they make up 22% of the neurons labelled"?

This has been clarified (line 164).

– The schematic in Figure S4 A implies that the ventral divergent interneuron in cervical cord is a double jump from the forelimb motor pool. It is likely that there is a 'mis-connection' in the schematic – the ventral neuron should synapse with the forelimb motor neurons, rather than the dorsal neuron to the ventral neuron.

We thank the reviewer for this comment and apologize. Indeed, we had made a mistake in the schematic of what is now Figure 6—figure supplement 1A. This has now been corrected.